# Symmetry Reveals Layerwise Dynamics:
# How Transformers Perform In-Context Classification

**Patrick Lutz** [1]   **Themistoklis Haris** [1]   **Arjun Chandra** [1]   **Aditya Gangrade** [1]   **Venkatesh Saligrama** [1]

## Abstract

Transformers can perform in-context classification from a few labeled examples, yet the inference-time algorithm remains opaque. We study multi-class linear classification in the hard no-margin regime and make the computation identifiable by enforcing feature- and label-permutation equivariance at every layer. This yields highly structured weights and enables interpretability while maintaining functional equivalence. From these models we extract an explicit depth-indexed recursion—an end-to-end identified, emergent update rule inside a softmax transformer, to our knowledge the first of its kind. Attention matrices formed from mixed feature–label Gram structure drive coupled updates of training points, labels, and the test probe. The resulting dynamics implement a geometry-driven algorithmic motif, which can provably amplify class separation and yields robust expected class alignment.

## 1 Introduction

Transformers trained for in-context learning (ICL) routinely solve learning problems using only a forward pass. Yet, the core mechanistic question remains open:

*What algorithm is the model running at inference time?*

A common hypothesis is that transformers implicitly simulate a generic optimizer—most often gradient descent—within the residual stream. In this paper, we show that for in-context classification, this is a misleading abstraction. Instead, we show that once natural task symmetries are enforced *layer by layer*, trained transformers instantiate a shared dynamical template: a **coupled feature–label algorithmic motif**. This motif is not a loose analogy: it yields a closed-form layerwise recursion (depth $\leftrightarrow$ iterations), sup-

ports theory with testable predictions, and reappears when we retrain transformers on classification tasks with different distributional and geometric structures.

**The obstacle: symmetry breaking hides the computation.** Classification problems possess inherent symmetries: permuting examples, permuting feature coordinates, or relabeling classes should not change the decision rule. While self-attention is invariant to token order, standard training does not enforce feature- or label-permutation symmetry. As a consequence, models achieve high accuracy, but encode spurious asymmetries in their weights. These asymmetries hinder interpretability by obscuring the underlying algorithmic structure. To move from models that predict well to *discovering the underlying algorithm*, we must break through this barrier. We do so by forcing the internal computations of the model to respect the task's symmetries.[2]

**The method: enforcing symmetry layer by layer.** We enforce symmetry by construction. At each layer, we conjugate the attention block with a random block permutation over feature and label coordinates (apply $P$, run attention, then apply $P^\top$). This forces each layer to implement the same computation in every symmetric coordinate system. We verify that the learned inference rule is preserved by checking output agreement and query/context sensitivity fingerprints. Empirically, behavior is unchanged; instead, the constraint acts as a structural denoiser, collapsing the learned weights onto a low-dimensional, interpretable manifold.

**Result 1: Coupled feature-label dynamics.** With symmetry enforced, the forward pass becomes algebraically readable: for linear multiclass classification we recover an explicit layerwise recursion (layer $\ell$ = iteration $\ell$). Let $\boldsymbol{X}_\ell \in \mathbb{R}^{n \times d}$ be the layer-$\ell$ feature vectors of the $n$ training tokens, and $\boldsymbol{Y}_\ell \in \mathbb{R}^{n \times K}$ their label vectors. Each iteration first builds an affinity between training tokens by combining feature similarity and label agreement,

$$\boldsymbol{A}_\ell = \mathrm{softmax}\big(\alpha \boldsymbol{X}_\ell \boldsymbol{X}_\ell^\top + \gamma \boldsymbol{Y}_\ell \boldsymbol{Y}_\ell^\top\big),$$

and then applies this same weighting to propagate both

---

[1]Boston University, Departments of Computer Science and Electrical & Computer Engineering. Correspondence to: Patrick Lutz <plutz@bu.edu>.

*Proceedings of the 43rd International Conference on Machine Learning*, Seoul, South Korea. PMLR 306, 2026. Copyright 2026 by the author(s).

[2]Code for reproducing our main results is available here: https://github.com/LutzPatrick/ICL-Symmetry

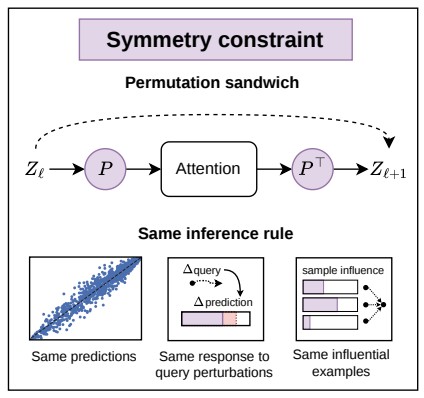
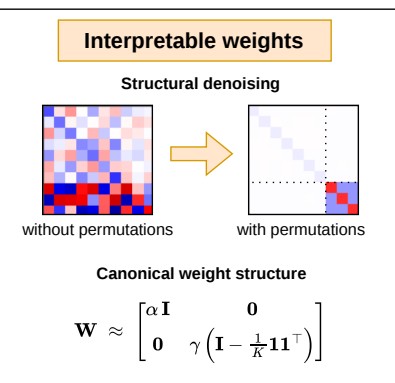
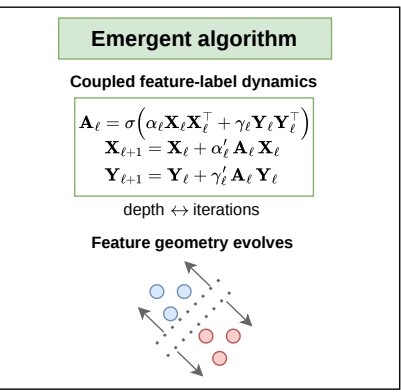

*Figure 1.* **Symmetry reveals coupled mean-shift inference in transformers.** *Left:* Enforcing feature/label permutation symmetry layer by layer preserves the inference rule: we match predictions, response to query perturbations (local decision rule), and influential examples (context sensitivity). *Middle:* The symmetry constraint denoises weights into a canonical low-dimensional structure. *Right:* The resulting structure makes the forward pass readable as a closed-form iterative classifier (depth $\leftrightarrow$ iterations) in which feature and label geometry co-evolve, yielding a geometry-driven algorithmic motif in which features and labels co-evolve.

training features and labels,

$$X_{\ell+1} = X_\ell + \alpha' A_\ell X_\ell, \quad Y_{\ell+1} = Y_\ell + \gamma' A_\ell Y_\ell.$$

The query token is updated analogously by attending to the training set, accumulating class evidence across iterations. Unlike GD-style ICL abstractions that update a classifier on fixed features, this is a *representation-shaping* procedure in which feature and label geometry co-evolve. We refer to this template as a coupled feature-label dynamic.

**Why this matters: Predictions, Theory, and Robustness.** With a closed-form layerwise recursion in hand, we can analyze the transformer's inference-time computation and derive testable predictions about behavior and performance. In a label-driven regime, the dynamics preserve class structure while monotonically amplifying the margin between class clusters. This yields a simple mechanism: a small number of labeled points anchors the classes, after which performance is driven primarily by geometric separation. A direct corollary is semi-supervised ICL: adding unlabeled points sharpens the geometry and improves accuracy even at a fixed label budget. More broadly, because the mechanism is geometry-driven, it should extend beyond linear separators when clusters are well separated, which we test in prototype (Voronoi cell) classification settings.

**Result 2: Dynamics predict behavior across tasks.** Our analysis predicts that the same coupled feature-label dynamic should succeed in semi-supervised, label-noise, and prototype (Voronoi) settings. Strikingly, when we retrain transformers on these tasks, this prediction carries over to the learned models: the trained transformers repeatedly recover the same symmetry-aligned weight structure and implement the same underlying dynamics (with task-specific parameter schedules). In other words, the closed-form iteration is not just an explanatory fit *after* training—it *predicts* the inference-time mechanism that re-emerges across tasks.

**Contributions.** In summary, we make three contributions:

1. **Symmetry as an identifiability protocol.** We enforce feature/label symmetries layer by layer and verify the inference rule is preserved (output alignment and query/context sensitivity), removing an important interpretability barrier and yielding structured, interpretable weights.

2. **Closed-form inference dynamics.** We extract a concise layerwise recursion for in-context classification (depth $\leftrightarrow$ iterations): a *coupled update dynamic* in which feature and label geometry co-evolve.

3. **A Shared Algorithmic Motif.** The extracted dynamics allows for theory and testable predictions that hold beyond the base setting; retraining transformers on semi-supervised, label-noise, and prototype tasks recovers the same weight structure and the same coupled algorithmic motif.

## 1.1 Related Work

**ICL as algorithm learning and implicit optimization.** Work in stylized in-context learning (ICL) settings trains transformers on sequences of examples sampled from a task family and analyzes the resulting inference-time procedure on held-out tasks (Garg et al., 2022; Zhang et al., 2024; Ahuja & Lopez-Paz, 2023). Several papers argue that, in linear regression and related regimes, attention and residual connections can realize closed-form estimators like gradient-descent updates (von Oswald et al., 2023; Akyürek et al., 2023; Dai et al., 2023). Recently, (Lutz et al., 2025) proposed a different mechanism based on the data representations of a trained transformer, bypassing the parametric perspectives of these earlier studies. Complementary theory formalizes this viewpoint as *algorithm learning* and studies learnability, generalization, and stability of the learned

solver (Li et al., 2023; Wies et al., 2023). At the same time, empirical correspondences between ICL and explicit gradient descent can be evaluation-dependent (Deutch et al., 2024). Our work adopts the algorithmic lens but focuses on multiclass classification in a hard regime; after enforcing the task symmetries layer-by-layer, the extracted computation is best described as a representation-space dynamical system rather than parameter-space empirical risk minimization.

**Symmetry and equivariance.** Symmetry-respecting architectures are a long-standing theme, including permutation-invariant set models (Zaheer et al., 2017; Lee et al., 2019) and more general group-equivariant networks that impose structured parameter sharing (Cohen & Welling, 2016; Maron et al., 2019; Satorras et al., 2021). Recent work studies permutation equivariance properties within transformer architectures and their consequences (Xu et al., 2024). We use equivariance differently: we enforce feature- and label-permutation symmetry *inside* each layer (a permutation-conjugation "sandwich"), not primarily to improve accuracy but to remove symmetry-breaking degrees of freedom that obscure identifiability, enabling closed-form extraction and cross-task motif comparisons.

**Mean-shift, diffusion, and semi-supervised learning.** Mean-shift is a classical mode-seeking and clustering method based on iterative kernel-weighted averaging (Comaniciu & Meer, 2002). Graph diffusion and label propagation methods likewise exploit unlabeled geometry by iterating a stochastic similarity operator (Zhu & Ghahramani, 2002). Our extracted recursion is naturally situated in this family: attention produces a learned stochastic similarity matrix and depth implements repeated diffusion steps, with label-dependent terms coupling representation updates to class information. This connection clarifies why behavior can be primarily margin-driven (geometry-first) and why similar dynamical templates can remain effective across perturbations that preserve the underlying symmetry structure. Related theoretical work studies self-attention as an interacting particle system and proves clustering phenomena consistent with mean-shift-like dynamics in deep attention stacks (Geshkovski et al., 2023; Rigollet, 2025).

**Further discussion.** Appx. A provides additional context on (i) how ICL emerges from pretraining distributions; (ii) Bayesian/kernel interpretations; and (iii) mechanistic circuit analyses.

## 2 Setup

**Supervised Prediction Task and In-Context Prediction.** Let $\mathcal{F}$ be a class of functions $f : \mathcal{X} \to \mathcal{Y}$ from features to labels. We study prediction problems where, given examples, the value of a map in $\mathcal{F}$ at a test point must be predicted. Specifically, given $n$ pairs $(\boldsymbol{x}_i, \boldsymbol{y}_i)_{i=1}^n$ with $\boldsymbol{y}_i = f(\boldsymbol{x}_i)$ for

some unknown $f \in \mathcal{F}$, and a test point $\boldsymbol{x}_{\text{test}}$, we would like to approximate $f(\boldsymbol{x}_{\text{test}})$. We call this the *prediction task induced by* $\mathcal{F}$. We set $\mathcal{X} = \mathbb{R}^d$ and $\mathcal{Y} = \mathbb{R}^K$ throughout.

*In-Context Prediction.* In in-context learning, the data $((\boldsymbol{x}_i, \boldsymbol{y}_i)_{i=1}^n, \boldsymbol{x}_{\text{test}})$ is formatted to a *prompt* $p$ that is fed a transformer. We use $\mathcal{P}$ to denote the set of prompts induced by $\mathcal{F}$, and we say that a prompt is *induced by* $f \in \mathcal{F}$ if for all $i$, $\boldsymbol{y}_i = f(\boldsymbol{x}_i)$. A *prediction function* is a map $\mathcal{P} \to \mathcal{Y}$, and the weights of the transformer encode a specific implementation of one such function. We will use $\boldsymbol{x}_{\text{test}}(p)$ as notation for the test point in $p$.

**Semi-Supervised Prediction and ICL.** Later in the paper, we also investigate semi-supervised prediction (SSP). This augments $n_{\text{lab}}$ labeled examples of the prediction task with $n_{\text{unlab}}$ unlabeled points $(\boldsymbol{x}_j)_{j=1}^{n_u}$, and demands the same solution. In semi-supervised ICL, the prompts are expanded to add these unlabeled examples coupled with a 'null' label.

**Transformers.** Following common conventions in the ICL literature (von Oswald et al., 2023; Zhang et al., 2024), we encode the input data (i.e., the prompt)

$$\tilde{\boldsymbol{X}}_0 = \begin{bmatrix} \boldsymbol{x}_1 & \ldots & \boldsymbol{x}_n & \boldsymbol{x}_{\text{test}} \end{bmatrix}^\top \in \mathbb{R}^{(n+1) \times d}$$
$$\tilde{\boldsymbol{Y}}_0 = \begin{bmatrix} \boldsymbol{y}_1 & \ldots & \boldsymbol{y}_n & \boldsymbol{0}_K \end{bmatrix}^\top \in \mathbb{R}^{(n+1) \times K}. \quad (1)$$
$$\mathbf{Z}_0 = \begin{bmatrix} \tilde{\boldsymbol{X}}_0 & \tilde{\boldsymbol{Y}}_0 \end{bmatrix} \in \mathbb{R}^{(n+1) \times (d+K)}.$$

We consider single-head, attention-only transformers (Vaswani et al., 2017), which transform an input $\mathbf{Z}_0 \in \mathbb{R}^{(n+1) \times (d+K)}$ over $L$ layers via

$$\mathbf{Q}_\ell = \mathbf{Z}_\ell \mathbf{W}_{Q,\ell}, \ \ \mathbf{K}_\ell = \mathbf{Z}_\ell \mathbf{W}_{K,\ell}, \ \ \mathbf{V}_\ell = \mathbf{Z}_\ell \mathbf{W}_{V,\ell},$$
$$\text{Attn}(\mathbf{Z}_\ell) = \sigma \left( \frac{\mathbf{Q}_\ell \mathbf{K}_\ell^\top}{\sqrt{d+K}} + \mathbf{M} \right) \mathbf{V}_\ell \mathbf{W}_{P,\ell}, \quad (2)$$
$$\boldsymbol{Z}_{\ell+1} = \mathbf{Z}_\ell + \text{Attn}(\mathbf{Z}_\ell),$$

where $\mathbf{M} = \begin{bmatrix} \mathbf{0}_{n+1} \mathbf{0}_n^\top & -\infty \mathbf{1}_{n+1} \end{bmatrix}$ is an additive masking bias that reproduces the causal constraint that earlier tokens cannot use query information and $\sigma$ is the row-wise softmax operator whereby for a matrix $\boldsymbol{A} = (A_{ij})_{i,j \in [n+1]}$ we have

$$(\sigma(\boldsymbol{A}))_{ij} = \exp(A_{ij}) / \sum_m \exp(A_{im}). \quad (3)$$

**Architectural scope.** End-to-end algorithms have so far mainly been identified for linear transformer models (von Oswald et al., 2023; Ahn et al., 2023). We take the next step by studying attention-only transformers with softmax, which capture data-dependent weighting while remaining tractable. To keep the mechanism identifiable, we omit MLPs, layer norm, and multi-head attention, and use this controlled setting to isolate how attention couples feature geometry and labels across layers.

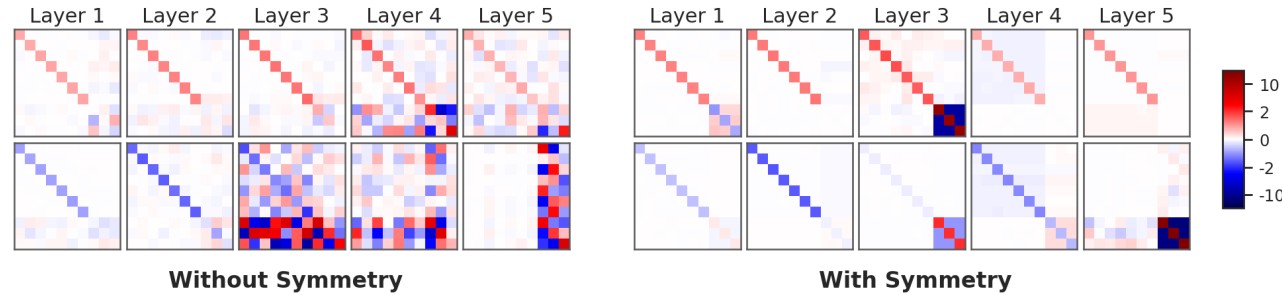

*Figure 2.* Learned weight matrices for the unconstrained transformer *(left)* and the symmetry-preserving transformer *(right)* trained on in-context linear classification. The unconstrained model exhibits little visible structure, whereas enforcing the task symmetries produces a more regular pattern that is easier to interpret. The top and bottom rows show $\boldsymbol{W}_{QK,\ell}$ and $\boldsymbol{W}_{VP,\ell}$, respectively.

# 3  Algorithm Extraction in In-Context Learning via Symmetries

In this section, we (i) formalize the natural symmetries that prediction tasks satify, and (ii) describe how we use these symmetries as a diagnostic tool to uncover the transformer's test-time inference procedure. Concretely, we do this by enforcing the symmetries *inside* the network at every layer, thus improving interpretability.

We begin by formally setting up a notion of symmetries of a prediction task.

**Definition 1.** *Let $G$ be a set of transformations $g : \mathcal{P} \to \mathcal{P}$. For $p \in \mathcal{P}$ induced by $f \in \mathcal{F}$ and $p' = g(p)$, let $f' \in \mathcal{F}$ denote any function inducing $p'$. We say $G$ captures* prediction invariance *if for all such $(p, f, p', f')$,*

$$f'(\boldsymbol{x}_{test}(p')) = f(\boldsymbol{x}_{test}(p)).$$

*If moreover the action $g$ extends to $\mathcal{Y}$, we say $G$ captures* prediction equivariance *if for all such $(p, f, p', f')$,*

$$f'(\boldsymbol{x}_{test}(p')) = g\big(f(\boldsymbol{x}_{test}(p))\big).$$

In words, such transformations $g$ modify the prompt in a structured way. If we apply $g$, the correct answer for the transformed prompt is either unchanged (invariance) or changes in the corresponding structured way (equivariance). Symmetries commonly arising in common ICL task families (e.g., regression/classification) include:

• *Feature symmetries.* The transformations $g$ apply an identical permutation to the feature coordinates of every $\boldsymbol{x}_i$ in the prompt and to the test point. This only relabels the coordinates, so the correct test prediction is unchanged.

• *Label symmetries.* The transformations $g$ relabel target classes by a fixed permutation (e.g., $0 \mapsto 1$ and $1 \mapsto 0$) applied to every $\boldsymbol{y}_i$ in the prompt. This only renames the labels, so the correct test label is permuted in the same way.

• *Permuting examples.* The transformations $g$ reorder the context pairs $(\boldsymbol{x}_i, \boldsymbol{y}_i)$. This does not change the underlying task, so the correct test prediction is unchanged.

*Example.* In linear regression with $f(x) = \langle w, x \rangle$, permuting the feature coordinates of all context inputs and the query by the same permutation $P$ gives a transformed prompt induced by $f'(x) = \langle Pw, x \rangle$. Hence $f'(Px_{\text{test}}) = f(x_{\text{test}})$, so this coordinate relabeling is a prediction invariance.

**Role of Symmetries in Prediction.** Symmetries transform the prompt without changing the underlying prediction task. It is therefore natural to ask whether a predictor respects these symmetries, i.e., whether applying $g$ to the prompt leaves its prediction unchanged (invariance) or transforms it by $g$ (equivariance). For example, empirical risk minimization respects them provided $g$ preserves the model class and the loss: applying $g$ maps any loss-minimizer for $p$ to a corresponding loss-minimizer for $g(p)$.

**Why symmetries help interpretability.** A trained transformer can implement a specific function via many distinct internal computations. Even if the *target algorithm* is equivariant, the network may implement it using intermediate representations defined in arbitrary, rotated coordinate systems. Consequently, the same input-output behavior can arise from obscure hidden-state dynamics whose representations do not reflect the problem's natural symmetries.

*Example.* Gradient descent (GD) for standard linear models is permutation equivariant: permuting the input feature coordinates simply results in a corresponding permutation of the weight updates. However, a transformer could simulate this algorithm while encoding the iterates in a basis that rotates layer-by-layer. While the final predictions remain correct (and equivariant), the internal activations would appear disordered, effectively masking the simple update structure of the simulated algorithm.

**Enforcing symmetry via randomization.** To resolve the ambiguity of arbitrary internal coordinates, we explicitly enforce symmetries layer-by-layer. We introduce a stochastic permutation operation inside each attention block. Crucially, if we applied a *fixed* permutation $P$, the transformation would be vacuous: it would act as a mere coordinate reparameterization (gauge choice) that the layer's linear weights

*Figure 3.* **Transformer symmetrization preserves the learned algorithm.** *Left:* The symmetrized transformer (S) matches the unconstrained (U): on the same prompt, U–S fingerprint matches the U–U baseline across query (local decision rule at the test point), context (which demonstrations drive the logits), and predictions, while the different-prompt control is near zero (mean over 2 048 tasks; averaged over 10 training runs; ±1 s.d. across runs). *Right:* After averaging predictions across 5 training runs, U and S are functionally identical: ground-truth-class probabilities agree with $R^2 > 0.98$ over 10 000 tasks, yielding the same decision boundaries.

could algebraically absorb. However, by *randomizing $P_\ell$* at every forward pass, we convert this gauge freedom into an active constraint. This prevents the model from relying on fixed coordinate choices and restricts the hypothesis class to functions that are permutation-equivariant:

$$\mathbf{Z}_{\ell+1} = \mathbf{Z}_\ell + \text{Attn}(\mathbf{Z}_\ell \mathbf{P}_\ell)\, \mathbf{P}_\ell^\top; \mathbf{P}_\ell \sim \text{Unif}(\mathfrak{S}_d \times \mathfrak{S}_k). \quad (4)$$

Here $\mathbf{Z}_\ell$ is the representation matrix. $\mathbf{P}_\ell$ independently permutes the subspaces corresponding to the $d$ input features and $k$ output dimensions. Because the permutation changes constantly, the attention block is forced to process information based on relationships rather than absolute positions. Figure 2 gives a visual indication of the effect: after symmetry enforcement, the learned transformer weights become substantially more structured.

**Interpretation via behavioral alignment.** We validate this intervention by comparing the original transformer to its symmetrized counterpart. Crucially, the two models remain closely aligned: on identical prompts, the unconstrained (U) and symmetrized (S) transformers match in their input-sensitivity and prediction fingerprints, while different-prompt controls do not (Figure 3). This indicates that symmetry enforcement preserves the learned strategy while removing arbitrary coordinate choices. The symmetrized network therefore serves as a *faithful proxy*: it implements the same underlying algorithm in a more canonical form suitable for reverse engineering.

## 4 Case Study: Linear Classification

In this section, we study transformers trained for in-context multi-class linear classification through the lens of symmetries. We apply the symmetry-based interpretability probe in Section 4.1 to extract the transformer's underlying update rule. We then study the emerging class of classifiers in Section 4.2: we prove convergence in the *label-dominated* regime, validate the mechanism in simulation, and discuss how the same classifier can extend to label noise, semi-supervised prompts, and selected non-linear tasks.

**Problem setup.** Each ICL instance is defined by $K$ latent unit-norm vectors $\boldsymbol{w}_1, \ldots, \boldsymbol{w}_K \in \mathbb{R}^d$. Given $n+1$ feature vectors $\boldsymbol{x}_1, \ldots, \boldsymbol{x}_n, \boldsymbol{x}_{\text{test}} \in \mathbb{R}^d$, we assign labels by

$$c_i = \arg \max_{k \in [K]} \langle \boldsymbol{w}_k, \boldsymbol{x}_i \rangle, \text{ and}$$
$$\boldsymbol{y}_i = \boldsymbol{e}_{c_i} \in \{0, 1\}^K. \quad (5)$$

where $\boldsymbol{e}_1, \ldots, \boldsymbol{e}_k$ are the standard basis vectors in $\mathbb{R}^K$, so $\boldsymbol{y}_i$ is the one-hot embedding of class $c_i$. In words, each $\boldsymbol{w}_k$ is a class-specific direction, and $\boldsymbol{x}_i$ is assigned to the class whose direction is most aligned with it.

Given the prompt built from examples $(\boldsymbol{x}_i, \boldsymbol{y}_i)_{i \in [n]}$ and a test feature $\boldsymbol{x}_{\text{test}}$, the goal is to predict $\boldsymbol{y}_{\text{test}}$, i.e., determine which class direction is most aligned with $\boldsymbol{x}_{\text{test}}$, without directly observing the $\{\boldsymbol{w}_k\}$. We denote $s(p) \in \mathbb{R}^k$ the transformer's predicted logit vector for the query point.

**Transformer training.** We train the transformer architecture (2) on i.i.d. instances with $\boldsymbol{x}_i \sim \mathcal{N}(\mathbf{0}, \boldsymbol{I}_d)$ and each $\boldsymbol{w}_k \sim \text{Unif}(\mathbb{S}^{d-1})$, where $\mathbb{S}^{d-1}$ is the unit sphere in $\mathbb{R}^d$. Because $\{\boldsymbol{w}_k\}$ vary across instances and are never revealed, achieving low loss requires the model to infer the induced linear decision rules from the in-context examples. See Appx. B.1 for additional training detail.

### 4.1 Extracting the Emergent Algorithm

**Step 1: Train a transformer.** We first verify that the unconstrained architecture in (2) solves the task. We fix $K = 3$, $d = 7$ and $n = 64$, and train transformers with depths $L \in \{1, 2, 5\}$. Figure 4 compares their performance to logistic regression and linear SVM across context lengths. The transformer is consistently competitive and improves with depth, indicating that solving this task requires $L > 1$ layers.

**Step 2: Make the weights easier to interpret.** Figure 2 compares a standard transformer trained without additional constraints (2) to a symmetry-enforced transformer trained under our approach (4). The symmetry-enforced model

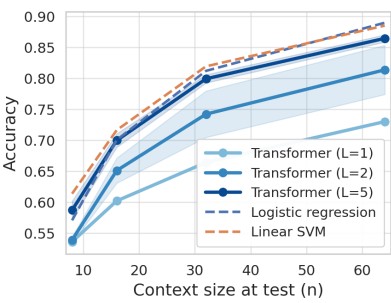

*Figure 4.* **Transformers match baselines.** Test performance vs. context length (mean over 10,000 tasks; averaged over 5 training runs; $\pm 1$ s.d. across runs).

exhibits a markedly more regular weight structure, which makes the learned computation easier to interpret.

**Behavioral alignment.** Crucially, symmetrization changes the *parameterization*, not the learned *algorithm*: it removes redundant degrees of freedom while preserving the model's input–output map and local decision rule. We test this by comparing unconstrained (U) and symmetrized (S) transformers with three "fingerprints," in the spirit of output- and gradient-level matching (e.g., Akyürek et al., 2023; von Oswald et al., 2023). We compare (i) the predicted ground-truth probability $\text{softmax}(s(p))_{c_{\text{test}}}$, (ii) the query Jacobian $\nabla_{\boldsymbol{x}_{\text{test}}} s(p) \in \mathbb{R}^{K \times d}$ capturing the local decision rule around the test input, and (iii) per-example context influence scores $S \in \mathbb{R}^{K \times n}$ where $S_{c,i} = \|\nabla_{\boldsymbol{x}_i} s_c(p)\|_2$ indicating which demonstrations drive the logits. See Appx. B.2 for details.

Our experiments show that symmetrization preserves the learned algorithm: it leaves not only the predictions, but also the local decision rule and context usage essentially unchanged. In Fig. 3, on the *same* prompt, U–S is as similar as U–U: the query Jacobian and context-influence finger-prints align strongly (Spearman's correlation $> 0.95$ and $\approx 0.8$), whereas the different-prompt control has near-zero alignment. Predicted probabilities also match instance-by-instance ($R^2 > 0.98$ after averaging over training seeds; Fig. 3, right). Further, additional experiments show that, on the aggregate level, overall test accuracy agrees across a broad sweep of problem instances (varying dimension, sample size, and margin; see Appx. B.2).

**Step 3: Extract the algorithm.** Using the structured weights from Step 2, we project each learned matrix onto a two-parameter block family matching the pattern in Fig. 2. Concretely, we approximate

$$\boldsymbol{W} \approx \begin{bmatrix} \alpha \, \boldsymbol{I}_d & \boldsymbol{0}_{d \times K} \\ \boldsymbol{0}_{K \times d} & \gamma \left( \boldsymbol{I}_K - \frac{1}{K} \boldsymbol{1}_K \boldsymbol{1}_K^\top \right) \end{bmatrix}. \quad (6)$$

This abstraction further preserves the model's implemented inference behavior, as quantified in Appendix B.2. For layer $\ell$, we denote the parameters of $\boldsymbol{W}_{\text{QK},\ell}$ by $(\alpha_\ell, \gamma_\ell)$ and those of $\boldsymbol{W}_{\text{VP},\ell}$ by $(\alpha'_\ell, \gamma'_\ell)$.

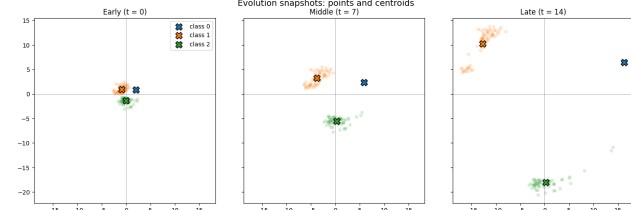

*Figure 5.* **Label-driven mean-shift increases class separation.** *Left:* Simulated dynamics with fixed parameters $(\alpha, \gamma, \alpha', \gamma') = (1, 5, 0.08, 0.1)$, showing class centroids drifting apart.

Weights of the form (6) commute with the layerwise per-mutations in the symmetrized architecture, so the permutation sandwich leaves the computation unchanged. Plugging this weight abstraction into the transformer update (2) then yields the two-line dynamics (see Appx. B.2 for detail)

$$\begin{aligned} \tilde{\boldsymbol{X}}_{\ell+1} &= \tilde{\boldsymbol{X}}_\ell + \alpha'_\ell \, \boldsymbol{A}_\ell \boldsymbol{X}_\ell, \\ \tilde{\boldsymbol{Y}}_{\ell+1} &= \tilde{\boldsymbol{Y}}_\ell + \gamma'_\ell \boldsymbol{A}_\ell \boldsymbol{Y}^c_\ell, \end{aligned} \quad (7)$$

where the attention from all tokens to the training tokens is

$$\boldsymbol{A}_\ell = \sigma\Big( \alpha_\ell \tilde{\boldsymbol{X}}_\ell \boldsymbol{X}_\ell^\top + \gamma_\ell \tilde{\boldsymbol{Y}}^c_\ell (\boldsymbol{Y}^c_\ell)^\top \Big) \in \mathbb{R}^{(n+1) \times n} \quad (8)$$

with $\sigma$ the row-wise softmax. Here, $\tilde{\boldsymbol{X}}, \tilde{\boldsymbol{Y}}$ contain all fea-tures and labels as defined in (1) and we set $\boldsymbol{X}_\ell \in \mathbb{R}^{n \times d}$ and $\boldsymbol{Y}_\ell \in \mathbb{R}^{n \times K}$ to denote the first $n$ rows (in-context examples only). Labels are centered across classes via

$$\boldsymbol{Y}^c_\ell = \boldsymbol{Y}_\ell \big( \boldsymbol{I} - \tfrac{1}{K} \boldsymbol{1}\boldsymbol{1}^\top \big) \quad \text{and} \quad \tilde{\boldsymbol{Y}}^c_\ell = \tilde{\boldsymbol{Y}}_\ell \big( \boldsymbol{I} - \tfrac{1}{K} \boldsymbol{1}\boldsymbol{1}^\top \big). \quad (9)$$

Thus $\boldsymbol{A}_\ell$ combines feature similarity and label agreement, and the same weights propagate both features and labels.

### 4.2 Analysis: The Coupled Mean-Shift Mechanism

In the previous section, we identified a *family* of attention-driven update rules (7) that the transformer instantiates. We now show that this family contains parameterizations whose induced dynamics implement a classification rule.

**Label-Driven Regime.** To make this concrete, we consider a fixed schedule $\alpha, \gamma, \alpha', \gamma'$ with positive parameters and analyze the corresponding iterations in the regime $\gamma/\alpha \gg 1$. In this regime, the attention matrices are dominated by the term $\gamma \tilde{\boldsymbol{Y}} \boldsymbol{Y}^\top$, and the resulting dynamics are effectively 'label-driven.'

**Connection to mean shift.** The resulting attention-driven recursion is reminiscent of the classical *mean-shift* proce-dure (Fukunaga & Hostetler, 1975). Mean-shift repeatedly moves each point toward a similarity-weighted local av-erage, which causes groups of mutually similar points to *collapse* to shared representatives (modes). The same con-traction appears here. Since each attention row is a prob-ability distribution, $\boldsymbol{A}_\ell \boldsymbol{X}_\ell$ is a weighted barycenter of the

context, and the update $\tilde{\boldsymbol{X}}_{\ell+1} = \tilde{\boldsymbol{X}}_\ell + \alpha'_\ell \boldsymbol{A}_\ell \boldsymbol{X}_\ell$ implements an averaging-style drift that merges clusters into prototypes. The crucial difference is that our similarity is *supervised*: $\boldsymbol{A}_\ell$ depends on both feature and label similarity via the $\tilde{\boldsymbol{Y}}$-term, so label-consistent points can attract and merge even when they are far in raw feature space. We therefore refer to this family as a *coupled mean-shift*: it imports the averaging-and-collapse geometry of classical mean shift into a supervised, label-aware classification setting.

**Classwise separation.** In the label-driven limit, attention over the context data becomes class-conditional: each point averages only with others sharing its label, giving a block-diagonal attention matrix and partitioning the data into class-wise clusters. This pushes class centroids apart in a way governed by the geometry of the class vectors $\{\boldsymbol{w}_k\}$, increasing the separability of the embeddings (Theorem C.1).

**Test Point and Prediction Dynamics.** While the training data undergoes the dynamics above, the initial movement of the test point $\boldsymbol{x}_{\text{test}}(\ell)$, and the prediction $\boldsymbol{y}_{\text{test}}(\ell)$ is not label-driven even in the above limit. This is because the initial $\boldsymbol{y}_{\text{test}}(0)$ is the zero vector, and so does not affect the "test-attention" $(A_\ell^{\text{test}})_i \propto \exp(\alpha \langle \boldsymbol{x}_{\text{test}}(\ell), \boldsymbol{x}_i(\ell) \rangle + \gamma \langle \boldsymbol{y}_{\text{test}}(\ell), \boldsymbol{y}_i(\ell) \rangle)_i$. Instead, this dynamics is dominated by the interaction of $\boldsymbol{x}_{\text{test}}(0)$ with the training datapoints. As our main theoretical result, we argue that under explicit pointwise train/test separation conditions, the dynamics of the test point drive $\boldsymbol{x}_{\text{test}}(\ell)$ towards the cluster of the *correct* label, and further drive its prediction $\boldsymbol{y}_{\text{test}}(\ell)$ towards this label.

**Theorem 4.1.** *Let $c^*$ be the correct class prediction for $\boldsymbol{x}_{\text{test}}$. Define the geometric alignments*

$$R_\ell := \min_{i: c_i = c^*} \langle \boldsymbol{x}_{\text{test}}(\ell), \boldsymbol{x}_i(\ell) \rangle, \text{ and}$$

$$L_\ell := \max_{i: c_i \neq c^*} \langle \boldsymbol{x}_{\text{test}}(\ell), \boldsymbol{x}_i(\ell) \rangle.$$

*Also define the test margin $\Delta_\ell = R_\ell - L_\ell$, the training margin $\Gamma_\ell$ as in Lemma 3, and the effective margin $\widetilde{\Delta}_\ell := \min(\Delta_\ell, \Gamma_\ell)$. If the class-clusters are well-separated in the pointwise sense of Lemma 3, $\widetilde{\Delta}_0 > 0$, and $\alpha \Delta_0 \geq \log(K) + \log(1 + 2/\widetilde{\Delta}_0)$, then*

*1. The geometric margin diverges as $\Delta_\ell \geq \Delta_0 (1 + \alpha')^\ell$.*
*2. Further, the* label *margin grows exponentially: For all $\ell \geq \lceil (\alpha')^{-1} \log \frac{\log(4K)}{\alpha \Delta_0} \rceil$,*

$$\forall c \neq c^*, \ (\boldsymbol{y}_{\text{test}}(\ell))_{c^*} - (\boldsymbol{y}_{\text{test}}(\ell))_c \geq (1 + \gamma')^\ell / 2.$$

In words, $(R_0, L_0)$ capture how well $\boldsymbol{x}_{\text{test}}$ aligns with points in the correct and incorrect classes, while $\Gamma_0$ captures the corresponding pointwise training separation. The effective quantity $\widetilde{\Delta}_0$ controls the assumption needed to keep the train/test recursion favorable. Notice that if even if $\Delta_0 > 0$ is arbitrarily small, a large enough $\alpha$ enables the condition

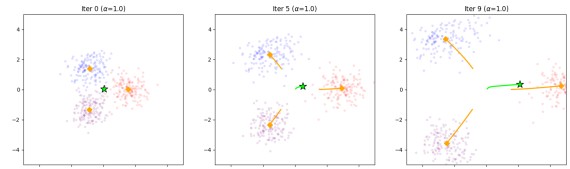

*Figure 6.* A simulation of the mean-shift trajectories with constant parameters $(\alpha, \gamma, \alpha'\gamma') = (1, 5, 0.08, 0.1)$. The test point 'follows' a trailing path towards its cluster.

above: thus, as for large $\alpha$, any initial bias in $\boldsymbol{x}_{\text{test}}$ towards the correct class-cluster is amplified exponentially through the geometry driven phase of this dynamics. Further, interpreting $(\boldsymbol{y}_{\text{test}}(\ell))_{c^*}$ as logits of the final prediction, after an initial burn in period determined by $\alpha'$, these are driven exponentially as well. Figure 6 illustrates this dynamics.

**Robustness to label noise.** The analysis suggests robustness to label noise because the recursion updates points using *averages* over many context tokens: correctly labeled points in a class reinforce one another, while randomly flipped labels are spread across incorrect classes, contribute mostly canceling noise, and have limited influence on the resulting class prototypes. Consistent with this, in Appendix B.3 we recover the same extracted iteration even when each label is independently flipped to a uniformly random incorrect class with 30% probability.

**Insights for Semi-Supervised Prediction.** The analysis reveals a simple mechanism: a small set of labeled points steers the class prototypes, while unlabeled (and test) points move by geometric alignment, effectively "latching onto" the appropriate cluster. As a result, if unlabeled points concentrate around the class clusters and at least a few labels are present to anchor them, the same dynamics can propagate those labels through the geometry and correctly classify the test point whenever it aligns with the right cluster. In Section 5, we study this semi-supervised regime in detail.

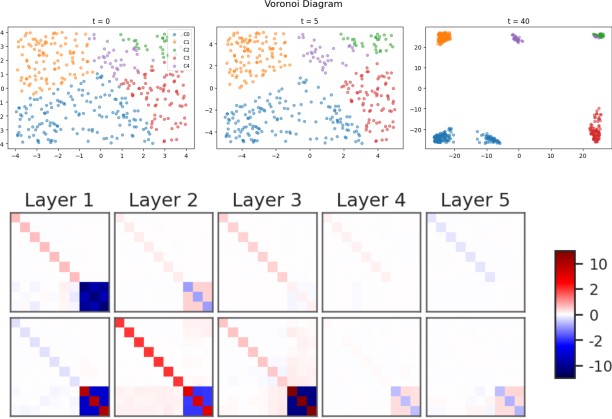

*Figure 7.* **Classification dynamics for Voronoi cells.** *Top:* Simulation for $(\alpha, \gamma, \alpha', \gamma') = (1, 5, 0.05, 0.2)$. *Bottom:* Trained transformer weights encode the same dynamics (7), matching Fig. 2.

**Nonlinear Classification.** Notice that in the label-driven regime, so long as the initial class-cluster means start out distinct, the dynamics will amplify their separation, *even if* the class distributions are anisotropic, or classes are separated by nonlinear boundaries. This in fact suggests that the same family of dynamics may be successful at classifying much richer problems than linear classification. We discuss the case of classifying into Voronoi cells in Section 4.3 below, and explore more settings in Appx. D.

### 4.3 Classification to Voronoi Cells

So far we have focused on the multiclass linear classification problem, where each class is defined by a fixed direction. We now consider nearest-centroid (Voronoi) classification where each class corresponds to a prototype, and the class-wise decision boundaries are piecewise linear.

**Setup.** We sample $K$ centroids $\{\mathbf{c}_k\}_{k=1}^K$ and $n$ feature vectors $\{\mathbf{x}_i\}_{i=1}^n$ independently from $\mathcal{N}(\mathbf{0}, \boldsymbol{I}_d)$ and assign labels by the nearest centroid,

$$y_i = \arg\min_{k \in [K]} \|\mathbf{x}_i - \mathbf{c}_k\|_2, \qquad (10)$$

which partitions the plane into $K$ Voronoi cells.

**Results.** Figure 7 (top) runs our extracted emergent dynamics on Voronoi data and shows that it correctly recovers the nearest-centroid clusters. This shows the mechanism is not specific to linear-score classifiers: the same geometry-driven dynamics extends to prototype-based, piecewise-linear tasks. Consistent with this, Figure 7 (bottom) shows that a symmetrized transformer trained on the Voronoi classification task learns the same family of update rules. Moreover, Appx. B.4 shows that the transformer achieves competitive performance on this task and strong alignment between the unconstrained and symmetrized transformers.

## 5 Transformers as Semi-Supervised In-Context Learners

We now move from fully supervised prompts to a semi-supervised setting: each prompt contains a small set of labeled examples and many additional *unlabeled* points (their label token is set to the all-zero vector). The question is whether a transformer can use these unlabeled points to improve classification.

Our analysis of the emergent classification dynamics motivates this question and yields a simple prediction: in the early, geometry-driven phase, the test point's behavior is controlled by how much it aligns with the correct versus incorrect clusters (the geometric margin $\Delta_\ell$ in Theorem 4.1). Adding unlabeled points from the same clusters sharpens this geometry, increasing the effective separation seen by the test point even without adding labels, and thereby makes

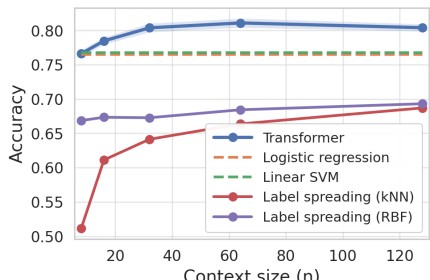

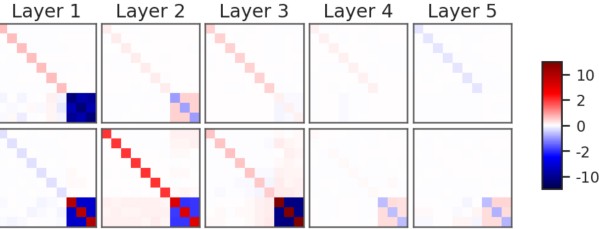

*Figure 8.* **Transformers use unlabeled context to improve semi-supervised in-context learning.** *Top:* Each prompt has 8 labeled examples; longer contexts add unlabeled points only. We plot accuracy vs. context length (mean over 10 000 tasks; averaged over 3 runs; ±1 s.d. across runs). *Bottom:* Trained transformer weights encode the same dynamics (7), matching Fig. 2.

the subsequent label-driven amplification more reliable.

**Setup.** We start from the same multi-class linear classification instance with examples $(x_i, y_i)_{i=1}^n$ and a query $x_{\text{test}}$. We construct a semi-supervised variant in two steps. First, we make the class structure more pronounced by shifting each feature vector slightly in the direction of its class:

$$\boldsymbol{x}_i \leftarrow \boldsymbol{x}_i + \eta\, \boldsymbol{w}_{c_i}, \qquad \eta > 0.$$

This increases the geometric separation between classes by moving points toward their class direction, strengthening within-class coherence and between-class margins. Second, we remove the labels for all but $n_{\text{lab}}$ examples, yielding a setup with $n_{\text{lab}}$ labeled and $n_{\text{unlab}} = n - n_{\text{lab}}$ unlabeled inputs.

**Benefit of Unlabeled Context (Fixed Label Budget).** To strictly isolate the model's ability to leverage unlabeled geometry, we perform a controlled experiment where $n_{\text{lab}} = 8$ is fixed to a scare budged, while $n_{\text{unlab}}$ varies in $\{0, 8, 24, 120\}$. We fix $K = 3$, $d = 7$, and $L = 5$, and train a single transformer with data shift parameter $\eta = 0.5$. With this choice of $\eta$, the classes are separable but the margin is small relative to the within-class spread, so the unlabeled geometry is only weakly informative and the SSL task is challenging.

Figure 8 presents the results. Standard supervised baselines (Logistic Regression, SVM; dashed lines) cannot utilize the unlabeled points, and show flat performance curves. In stark contrast, the Transformer (Blue) achieves significant accuracy gains as the context size $n$ grows (rising from $\sim 75\%$

to $> 80\%$). This confirms that the model is not merely smoothing over the labeled examples, but actively aggregating geometric information from the unlabeled tokens to refine its decision boundary. Moreover, the weights of these transformers again encode the same mean-shift mechanism as in Sec. 4.1 (Fig. 8, bottom), showing that this motif extends to SSL. Appendix B.5 provides further details and an ablation showing that the gains come from the unlabeled tokens themselves, not from extra tokens increasing the transformer's capacity or compute.

**Transformer vs. classical SSL.** Figure 8 highlights a sharp gap with standard graph-based SSL. Despite the geometric structure in the context, Label Spreading (purple/red) performs worse than simple supervised baselines. This phenomenon is consistent with known failure modes on high-dimensional Gaussian data where Euclidean neighborhoods are fragile without careful kernel tuning. In contrast, the transformer reliably uses the unlabeled points and outperforms these SSL baselines as $n$ grows, indicating that it has learned a robust way to aggregate geometric signal.

## 6 Conclusion

In this work, we make in-context classification more identifiable by enforcing the task's natural symmetries at every layer. This structure lets us extract an explicit attention-driven recursion from the transformer. Strong behavioral alignment shows that the extracted dynamics closely matches the transformer's actual test-time computation, giving a concrete characterization of its mechanism. We further show that this family of dynamics can implement a label-aware, mean-shift-like update that pulls points toward class prototypes. The same mechanism also appears in more complex settings, including Voronoi-cell classification and semi-supervised classification, where we uncover a surprising ability of in-context learning transformers to benefit from unlabeled context examples.

This analysis focuses on synthetic tasks with clear, explicit symmetries. While many practical applications exhibit analogous structure, systematically identifying and cataloging such symmetries remains an important direction for future work. Moreover, our method enforces symmetries layer by layer, making it particularly well suited to recovering permutation-commuting mechanisms. Extending this framework to in-context learning mechanisms that are not well characterized by layer-wise symmetries is therefore a natural next step. More broadly, understanding whether, when, and why transformers need to break symmetry to solve certain problems may provide further insight into the mechanisms underlying in-context learning.

## Acknowledgments

This research was supported by the National Science Foundation grant CPS-2317079.

## Impact Statement

This paper presents work whose goal is to advance the field of Machine Learning. There are many potential societal consequences of our work, none which we feel must be specifically highlighted here.

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

# A Additional Related Work

**Emergence of ICL from pretraining distributions.** Beyond controlled "train-from-scratch" studies, several works emphasize that the *pretraining data distribution* can strongly influence whether and how ICL emerges. In particular, properties such as task-family mixtures, long-range coherence, or "burstiness" can shape the implicit meta-learning behavior that appears at inference time (Chan et al., 2022; Xie et al., 2022; Yadlowsky et al., 2023). This line of work is complementary to ours: we hold the task distribution fixed and instead use symmetry to make the learned inference procedure identifiable and comparable across perturbations.

**Bayesian and kernel perspectives on ICL.** Several works interpret ICL as approximate Bayesian inference or Bayesian model averaging in latent-variable task models, providing statistical characterizations of the in-context predictor and its dependence on prompt length, depth, and pretraining distributions (Xie et al., 2022; Panwar et al., 2024; Zhang et al., 2023). A closely related view explains ICL behavior through kernel regression: attention weights act as similarity kernels, yielding predictors that resemble kernel smoothing and nearest-neighbor-like rules (Han et al., 2025). These perspectives are compatible with our findings at the level of *similarity-based* computation, but our extracted dynamics go beyond a single-shot kernel predictor: the model iteratively updates representations (and label-related state) across layers, producing a coupled diffusion/mean-shift process whose analysis yields concrete geometric predictions.

**Mechanistic interpretability and circuit analyses of ICL.** Mechanistic interpretability work has identified specific transformer circuits implicated in ICL, notably the emergence of induction heads as a mechanism for copying and pattern completion that correlates with ICL capability (Olsson et al., 2022). Survey work synthesizes broader mechanistic and theoretical hypotheses for ICL across settings and model scales (Zhou et al., 2024). Our approach is complementary: rather than isolating a local circuit, we use symmetry to make the *entire* forward-pass computation identifiable in a controlled classification setting, enabling closed-form extraction and dynamical analysis.

# B Supplemental material on Empirical Protocols

## B.1 Supplemental material on transformer training

Unless otherwise stated, all transformer experiments use $K = 3$ classes, feature dimension $d = 7$, and context length $n = 64$. We train with Adam at learning rate $10^{-3}$, $\beta_1 = 0.9$, $\beta_2 = 0.999$, weight decay 0, batch size $8192$, for 20,000 optimizer steps. We use no gradient clipping, dropout, label smoothing, or other regularization. The training loss is the cross-entropy between the ground-truth class of the query point and the transformer-predicted logits for that query.

Training prompts are sampled online at each optimization step rather than from a fixed finite dataset. Accordingly, training is best viewed as stochastic optimization of the expected loss under the task-generating distribution, with each minibatch providing a Monte Carlo estimate of the population objective.

All weight matrices $W \in \mathbb{R}^{(d+K) \times (d+K)}$ are initialized entrywise independently as $W_{ij} \sim \text{Unif}[-\frac{1}{d+K}, \frac{1}{d+K}]$. This uses a smaller initialization scale than standard Xavier-style schemes, which we found to improve training stability in our setup.

For the symmetrized model, during training and for each layer $\ell$ and batch element $b$, we sample independent random permutation matrices on the feature and label blocks, $P_{\ell,b}^{(x)} \in \mathbb{R}^{d \times d}$ and $P_{\ell,b}^{(y)} \in \mathbb{R}^{K \times K}$, and form

$$P_{\ell,b} = \text{diag}\big(P_{\ell,b}^{(x)}, P_{\ell,b}^{(y)}\big).$$

The layer update is then

$$Z_{\ell+1}^{(b)} = Z_\ell^{(b)} + \text{Attn}\Big(Z_\ell^{(b)} P_{\ell,b}\Big) P_{\ell,b}^\top.$$

Thus, attention is computed on a block-permuted representation and the resulting update is mapped back by the inverse permutation before the residual addition. Empirically, the learned weights approximately commute with these permutations, so the sandwich becomes unnecessary after training; we therefore apply symmetrization only during training and omit it at evaluation time. Random seeds are sampled independently, and we average over 5 runs unless otherwise specified.

## B.2 Supplemental material on algorithm extraction

**Influence and Function Alignment.** We evaluate whether Unconstrained (U) and Symmetrized (S) transformers implement the same *inference rule* by probing (i) *what in the prompt they rely on* and (ii) *what they output*. Concretely, we track

three signals, each computed from the predicts query-point logit vector $s(p) \in \mathbb{R}^K$:

**(1) Query gradient (local decision rule at test time).** We compute the gradient of each logit with respect to the *query* features,

$$\nabla_{x_{\text{test}}} s(p) \in \mathbb{R}^{K \times d}. \tag{11}$$

This probes the model's *local geometry*: two models with aligned query gradients are locally sensitive to the same directions in input space, i.e., they induce a similar local decision boundary around $x_{\text{test}}$.

**(2) Context-gradient sensitivity (influence over demonstrations).** For each context token $i \leq n - 1$ and class $k$, we measure how much the predicted logit changes under a small perturbation of that context example's features,

$$S_{c,i} \;=\; \big\|\nabla_{x_i}\big[s(p)\big]_c\big\|_2. \tag{12}$$

This yields a map in $S \in \mathbb{R}^{K \times n}$ that functions like an *influence / saliency map over the prompt*: high values indicate which demonstrations the model is locally using for which test point class logits.

**(3) Prediction discrepancy (functional agreement).** We compare outputs directly by looking at the probability assigned to the ground-truth test class,

$$\big[\text{softmax}(s(p))\big]_{c_{\text{test}}}, \tag{13}$$

and report the squared differences in $p_{\text{true}}$ across models. This is the most direct measure of *functional equivalence* (up to calibration).

For the two gradient-based probes (context sensitivity and query gradient), we flatten each per-prompt tensor and report agreement via Spearman and Pearson correlations. We report all three signals under three comparisons: (i) **U–S** (same prompt) is the main cross-family test of alignment; (ii) **U–U** (same prompt) measures training seed-to-seed variability within the unconstrained family; and (iii) **A–B** (same model) is a negative control using two independently sampled prompts, which should destroy prompt-specific influence structure (and increase prediction discrepancy), ruling out trivial correlations driven by architecture or preprocessing.

**Generalization across problem instances.** While the main text demonstrates algorithmic equivalence via output correlation and internal attention dynamics, we also performed an extensive sweep to ensure this holds across problem settings. Figure 9 confirms that the unconstrained and symmetrized models achieve indistinguishable accuracy curves across varying the number of layers $L \in [2, 16]$, context length $n \in [8, 40]$, data dimension $d \in [6, 18]$, and margin $\eta \in [0, 1]$. Both models track these task changes nearly identically, consistent with them implementing the same algorithmic behavior.

**Evidence that the weight abstraction is faithful.** Step 3 argues that a low-parameter, diagonal weight abstraction preserves the transformer's input–output behavior. We test this by progressively compressing the symmetrized model's weights and measuring alignment with the original, unconstrained transformer:

1. *Four clusters recover full accuracy.* Figure 10 (left) shows that, in the symmetrized model, clustering each weight matrix into $k = 4$ groups of equal coefficients already matches the model's accuracy. The clusters correspond to the four structural regions: the top-left block diagonal, the bottom-right block diagonal, the bottom-right background, and the zero background. Using this structure yields a three-parameter-per-layer abstraction,

$$\boldsymbol{W} \;\approx\; \begin{bmatrix} \alpha\,\boldsymbol{I}_d & \boldsymbol{0}_{d \times K} \\ \boldsymbol{0}_{K \times d} & \gamma\left(\boldsymbol{I}_K + \delta\boldsymbol{1}_K\boldsymbol{1}_K^\top\right) \end{bmatrix}, \tag{14}$$

    and Figure 10 (right) shows that its input–output alignment with the original transformer is essentially identical to that of the symmetrized model—i.e., the abstraction introduces no additional behavioral deviation beyond symmetrization.

2. *Two parameters per layer suffice.* Figure 11 shows that the transformer consistently learns a bottom-right background coefficient of approximately $\delta \approx -\frac{1}{K}$. We therefore fix this parameter and reduce the parameterization to two per layer which gives the abstraction used in the main text. Figure 10 (left) shows that this two-parameter model still closely matches the original transformer's outputs ($R^2 \approx 0.9$), with only a modest drop from $R^2 \approx 0.96$. This supports that the abstraction remains behaviorally faithful while being substantially simpler.

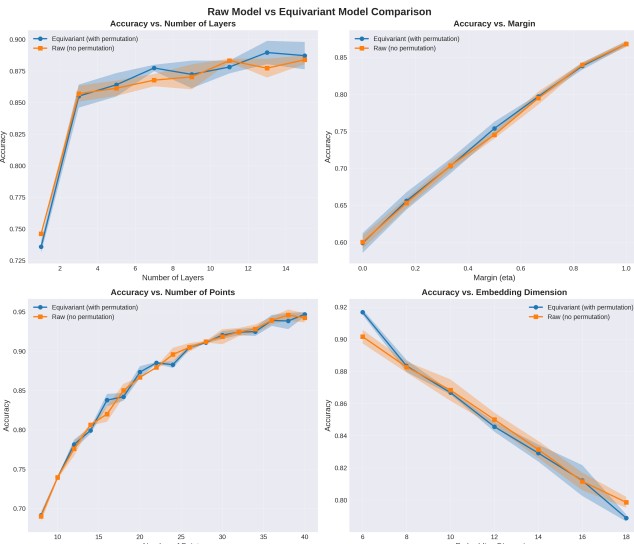

*Figure 9.* We compare predictive performance of unconstrained and symmetry-preserving transformers across problem instances (dimension, sample size, data margin) and model depths. Both architectures achieve the same performance throughout.

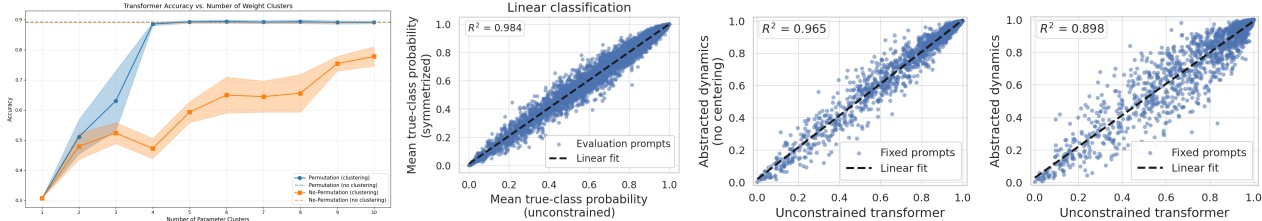

*Figure 10.* **Weight abstraction preserves the symmetric transformer's input–output behavior.** *Left:* Clustering the learned weight matrices (Figure 2) reveals a simple, low-dimensional structure. In the symmetry-preserving model, summarizing each matrix with just four parameters retains essentially the full predictive performance. *Right (three panels):* Alignment with the original unconstrained transformer for (left to right) the symmetrized model, the weight-abstracted dynamics with three layer-wise hyperparameters (no explicit label centering), and the further-simplified dynamics with two layer-wise hyperparameters (with explicit label centering).

**Developing transformer iteration under weight abstraction.** To pass from the transformer update (2) to the recursion (7), define

$$\boldsymbol{W}_{\mathrm{QK},\ell} := \frac{\boldsymbol{W}_{Q,\ell}\boldsymbol{W}_{K,\ell}^\top}{\sqrt{d+K}}, \qquad \boldsymbol{W}_{\mathrm{VP},\ell} := \boldsymbol{W}_{V,\ell}\boldsymbol{W}_{P,\ell}.$$

Writing $\boldsymbol{Z}_\ell = [\, \tilde{\boldsymbol{X}}_\ell \ \tilde{\boldsymbol{Y}}_\ell \,]$, the full attention score matrix is

$$\boldsymbol{Z}_\ell \boldsymbol{W}_{\mathrm{QK},\ell} \boldsymbol{Z}_\ell^\top + \boldsymbol{M}.$$

Under the block approximation (6), $\boldsymbol{W}_{\mathrm{QK},\ell}$ acts as $\alpha_\ell \boldsymbol{I}_d$ on features and as $\gamma_\ell(\boldsymbol{I}_K - \frac{1}{K}\mathbf{1}\mathbf{1}^\top)$ on labels, so

$$\boldsymbol{Z}_\ell \boldsymbol{W}_{\mathrm{QK},\ell} \boldsymbol{Z}_\ell^\top + \boldsymbol{M} \ \approx \ \alpha_\ell \tilde{\boldsymbol{X}}_\ell \tilde{\boldsymbol{X}}_\ell^\top + \gamma_\ell \tilde{\boldsymbol{Y}}_\ell^c (\tilde{\boldsymbol{Y}}_\ell^c)^\top + \boldsymbol{M}.$$

Since $\boldsymbol{M}$ sets the last column (the query token) to $-\infty$, the softmax assigns zero attention to that column. Thus attention is effectively taken only over the first $n$ training tokens, giving

$$\boldsymbol{A}_\ell = \sigma\Big(\alpha_\ell \tilde{\boldsymbol{X}}_\ell \boldsymbol{X}_\ell^\top + \gamma_\ell \tilde{\boldsymbol{Y}}_\ell^c (\boldsymbol{Y}_\ell^c)^\top\Big) \in \mathbb{R}^{(n+1)\times n}.$$

Likewise, the value/output map reduces to

$$\begin{bmatrix} \boldsymbol{X}_\ell & \boldsymbol{Y}_\ell \end{bmatrix} \boldsymbol{W}_{\mathrm{VP},\ell} \ \approx \ \begin{bmatrix} \alpha_\ell' \boldsymbol{X}_\ell & \gamma_\ell' \boldsymbol{Y}_\ell^c \end{bmatrix}.$$

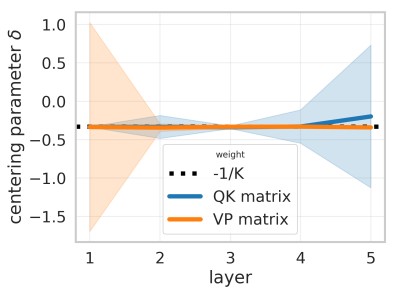 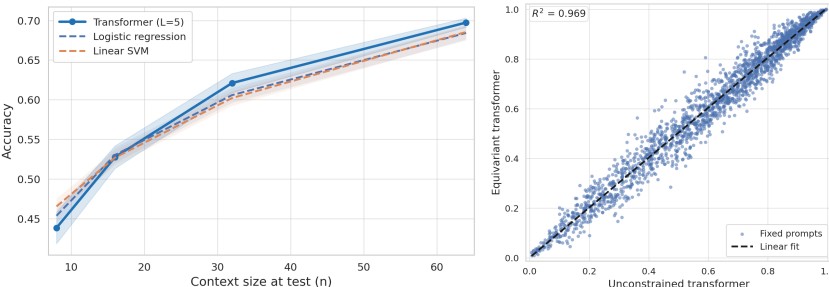

*Figure 11.* **Transformer learns label centering.** Parameter $\delta_\ell$ as in intermediary weight abstraction (1) motivates fixing $\delta = -\frac{1}{K}$. Median and std. in symmetrized transformer over 5 training runs.

*Figure 12.* Accuracy on the noisy linear classification task as a function of context size, comparing the transformer (mean over 3 runs, $L = 5$, trained with $n = 64$) to logistic regression and SVM. *Right:* Ground-truth-class probability for the query point: symmetrized transformer vs. unconstrained transformer on the same noisy linear classification task, mean across 3 training seeds are shown.

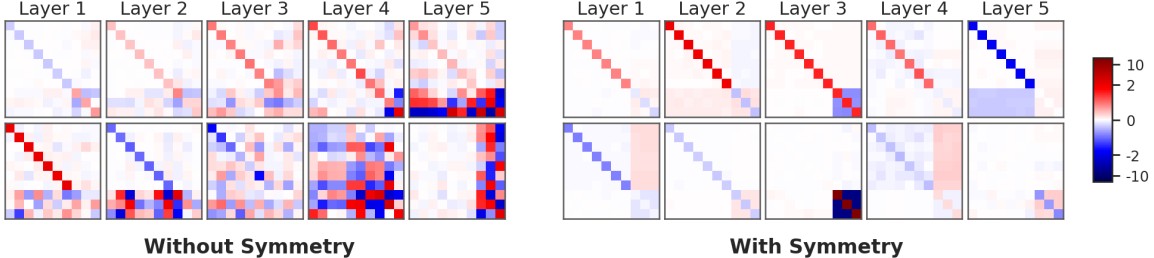

*Figure 13.* Learned weight matrices for the unconstrained transformer *(left)* and the symmetry-preserving transformer *(right)* trained on noisy linear classification. The unconstrained model exhibits little visible structure, whereas enforcing the task symmetries produces a more regular pattern that is easier to interpret. The top and bottom rows show $\boldsymbol{W}_{QK,\ell}$ and $\boldsymbol{W}_{VP,\ell}$, respectively.

Substituting into the residual update and separating feature and label blocks yields

$$\tilde{\boldsymbol{X}}_{\ell+1} = \tilde{\boldsymbol{X}}_\ell + \alpha'_\ell \boldsymbol{A}_\ell \boldsymbol{X}_\ell, \qquad \tilde{\boldsymbol{Y}}_{\ell+1} = \tilde{\boldsymbol{Y}}_\ell + \gamma'_\ell \boldsymbol{A}_\ell \boldsymbol{Y}_\ell^c,$$

which is exactly (7).

### B.3 Supplemental material on the robustness to noise

In the main text, we extracted a mean-shift-style classifier from transformers trained on our synthetic tasks. Here we show that the same qualitative mechanism persists under substantial label noise: each labeled example is independently flipped to a uniformly random incorrect class with probability $p = 0.3$.

Figure 12 (Left) reports the unconstrained transformer's accuracy under noise across context sizes, showing performance comparable to standard baselines. Figure 12 (Right) compares the predicted probability of the ground-truth class for unconstrained versus symmetrized models; the close agreement indicates that the symmetry projection does not change the learned predictor. Finally, Figure 13 visualizes the learned weights under label noise. The dominant structure that emerges closely matches the noiseless case: weights again cluster into the same diagonal groups with the same functional roles in the update. This supports the interpretation from the main text that the transformer is still implementing the same latent algorithm—an attention-induced, coupled mean-shift classifier—even when supervision is substantially corrupted.

### B.4 Supplementary material on the Voronoi cell problem

In the main text, we showed that the extracted mean-shift classifier can solve the Voronoi cell classification problem. Here we provide additional experimental details confirming that (i) a transformer trained end-to-end also solves this task reliably, and (ii) its behavior is consistent with the same mean-shift-style iteration we identified in the linear classification setting.

**Task construction.** Each ICL instance is defined by $K$ randomly sampled sites (centroids) $\{c_k\}_{k=1}^{K} \subset \mathbb{R}^d$. For each batch element we draw

$$c_k \sim \mathcal{N}(0, I_d), \qquad x_i \sim \mathcal{N}(0, I_d),$$

independently across $k$ and $i$. Each point is labeled by the nearest site in Euclidean distance,

$$y_i = \arg \min_{k \in [K]} \|x_i - c_k\|_2.$$

This induces a Voronoi partition of $\mathbb{R}^d$ with class regions determined jointly by the directions and magnitudes of the sites. For any pair of centroids, the tie set $\{x : \|x - \mathbf{c}_a\|_2 = \|x - \mathbf{c}_b\|_2\}$ is a hyperplane, so the classifier is a collection of linear boundaries. As before, the transformer input is a context of $n$ feature–label pairs $(x_i, \text{onehot}(y_i))$ and a query point $x_{\text{test}}$; the model is trained to predict the query label.

**Transformer performance and baselines.** Figure 14 (Left) reports accuracy for $K = 5$ classes as a function of context length, comparing an unconstrained transformer to a nearest-neighbor baseline. The transformer consistently exceeds the nearest-neighbor baseline, confirming that it learns a nontrivial in-context strategy for the classification problem.

**Agreement between unconstrained and symmetrized models.** Figure 14 (Right) compares the predicted class probabilities produced by the unconstrained and symmetrized transformers on the same test instances. The two models closely agree ($R^2 > 0.95$), indicating that enforcing the symmetry constraints does not change the learned solution, but instead exposes it more cleanly for analysis.

**Evidence for the same underlying algorithm.** Finally, Figure 15 visualizes the learned weights with and without symmetrization. The qualitative patterns mirror those observed in our linear classification experiments: the same cluster structure and the same functional roles of weight groups reappear. This supports the interpretation from the main text that the transformer implements essentially the same coupled mean-shift iteration, now operating on the geometry induced by nearest-centroid (Voronoi) labels.

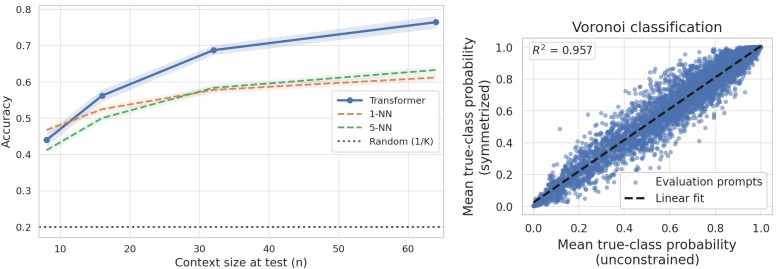

*Figure 14. Left:* Accuracy on the Voronoi (nearest-centroid) classification task as a function of context size, comparing the transformer (mean over 3 runs, $L = 5$, trained with $n = 64$) to nearest-neighbor baselines. *Right:* Ground-truth-class probability for the query point: symmetrized transformer vs. unconstrained transformer on the same Voronoi task, mean across 5 training seeds are shown.

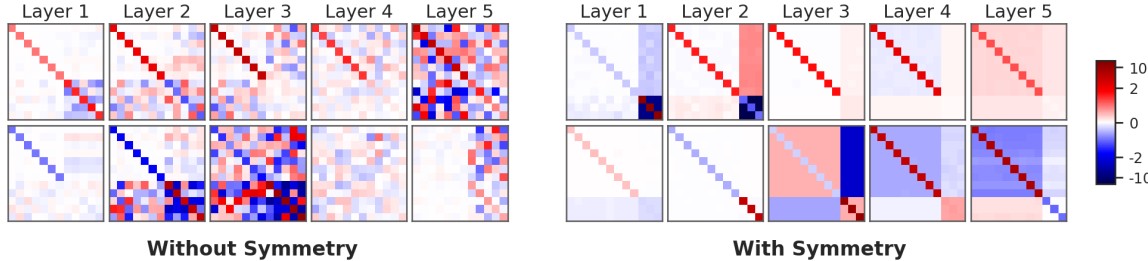

*Figure 15.* Learned weight matrices for the unconstrained transformer *(left)* and the symmetry-preserving transformer *(right)* trained on in-context Voronoi cell classification. The unconstrained model exhibits little visible structure, whereas enforcing the task symmetries produces a more regular pattern that is easier to interpret. The top and bottom rows show $\boldsymbol{W}_{QK,\ell}$ and $\boldsymbol{W}_{VP,\ell}$, respectively.

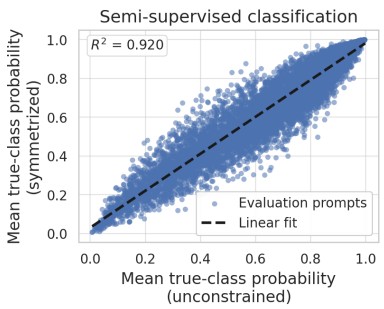 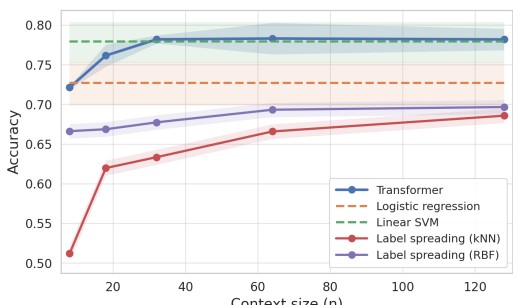

*Figure 16. Left:* Ground-truth query-class probability predicted by the symmetrized vs. unconstrained transformer on the same semi-supervised classification task (mean over 5 training seeds). *Right:* **Ablation with Unstructured Noise.** When unlabeled context points are replaced with random noise $\mathcal{N}(0, I_d)$, the Transformer (Blue) ceases to outperform the supervised Linear SVM baseline (Green dashed). Compare this to Figure 8, where the Transformer leverages geometric structure to exceed the supervised baseline by $> 10\%$. This confirms the model relies on the geometry of the unlabeled data to improve classification.

## B.5 Supplementary material on the semi-supervised classification setting

In the main text, we showed that the transformer uses unlabeled context points to improve query prediction under a fixed label budget. Here we show that it does so via the same coupled mean-shift iteration identified in the fully supervised setting.

Figure 16 (Left) compares the probability assigned to the ground-truth query class by the unconstrained and symmetrized transformers on the exact same inputs after averaging across 5 training seeds. The two predictions match closely ($R^2 = 0.92$) across a wide range of problem instances, indicating that the models are functionally aligned; symmetrization does not change the underlying computation, it simply makes it easier to expose and analyze.

Figure 17 provides complementary evidence at the parameter level: the learned weights exhibit the same cluster structure as in our linear classification experiments. Together, these results support the interpretation that the transformer implements essentially the same coupled mean-shift iteration in the semi-supervised regime, now operating jointly on the feature geometry and the partially observed labels.

**Semi-Supervised Baselines.** To evaluate geometric utilization (Figure 8), we compare the Transformer against **Label Spreading** (Zhou et al., 2003), a graph-based semi-supervised algorithm. We utilize the k-nearest neighbor (k-NN) kernel with $k = \max(5, \lfloor \sqrt{n} \rfloor)$. This kernel choice makes the baseline scale-invariant. The "Supervised" baseline models are a Logistic Regression and a linear SVM model trained only on the subset of labeled examples, ignoring the unlabeled tokens.

**Ablation: The Role of Geometric Structure.** To verify that the performance gains in Figure 8 stem strictly from geometric aggregation (the mean-shift mechanism) rather than architectural capacity, we conduct an ablation where the unlabeled context tokens are replaced with unstructured Gaussian noise, $\boldsymbol{z}_i \sim \mathcal{N}(0, I_d)$.

Figure 16 (Right) shows the results. In this unstructured setting, the Transformer (Blue line) no longer surpasses the supervised ceiling. Instead, it merely converges to the performance of the Linear SVM (Green dashed line). Crucially, the "super-supervised" lift observed Figure 8 (where accuracy exceeded $85\%$ vs. the SVM's $75\%$) completely vanishes. This confirms that the model is actively reading the manifold structure of the unlabeled data; when that structure is removed, the mean-shift mechanism defaults to a standard supervised solution, proving the gains are geometry-driven.

## B.6 Supplementary material on baseline implementations

We evaluate different baselines depending on the task setting. In this section, we give detailed information on the baseline methods we used in this work. We report accuracy, comparing the predicted query label to the true query label, and report a 95% Wilson confidence interval.

### B.6.1 FULLY SUPERVISED BASELINES.

In the fully supervised setting, we use inductive classifiers trained only on the labeled context points. Unlabeled context points and the query point are not used for training. The trained classifier is then evaluated on the query feature vector.

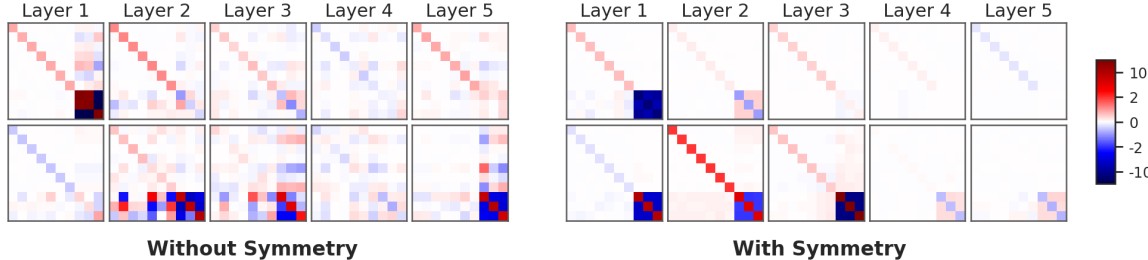

*Figure 17.* Learned weight matrices for the unconstrained transformer (left) and the symmetry-preserving transformer (right) trained on semi-supervised in-context linear classification. The unconstrained model exhibits little visible structure, whereas enforcing the task symmetries produces a more regular pattern that is easier to interpret. The top and bottom rows show $W_{QK,\ell}$ and $W_{VP,\ell}$, respectively.

**Logistic regression.** The first fully supervised baseline is logistic regression, implemented using `sklearn.linear_model.LogisticRegression`. The model is fit separately within each episode on the labeled context examples, using

$$\texttt{solver} = \texttt{lbfgs}, \qquad \texttt{max\_iter} = 400.$$

For each evaluated context size, the inverse regularization strength $C$ is selected on a validation set from the grid

$$C \in \{10^{-3}, 10^{-2}, 10^{-1}, 1, 10, 10^2, 10^3\}.$$

The predicted class is obtained directly from the fitted classifier. If the labeled context contains only one class, no model is fit; the baseline predicts that class.

**Linear SVM.** The second fully supervised baseline is a linear support vector machine, implemented using `sklearn.svm.LinearSVC`. The model is fit separately within each episode on the labeled context examples, using

$$\texttt{max\_iter} = 8000.$$

As for logistic regression, the regularization parameter $C$ is selected separately for each evaluated context size on a validation set from the grid

$$C \in \{10^{-3}, 10^{-2}, 10^{-1}, 1, 10, 10^2, 10^3\}.$$

The predicted class is obtained directly from the fitted SVM. If the labeled context contains only one class, no model is fit; the baseline predicts that class.

### B.6.2 SEMI-SUPERVISED BASELINES.

In the semi-supervised setting, we use transductive label-spreading baselines. These methods are fit on the full episode, including labeled context points, unlabeled context points, and the query point. Labeled context points are assigned their observed class labels, while unlabeled context points and the query are assigned label $-1$. Before fitting either semi-supervised method, features are standardized within each episode using

$$\texttt{StandardScaler(with\_mean=True, with\_std=True)}.$$

**kNN label spreading.** The first semi-supervised baseline is `LabelSpreading` with a k-nearest-neighbor graph. We use

$$\texttt{kernel} = \texttt{knn}, \qquad \alpha = 0.2, \qquad \texttt{max\_iter} = 2000, \qquad \texttt{tol} = 10^{-4}.$$

The number of neighbors is selected adaptively per episode as

$$k = \min\left(\text{clip}\left(\lceil\sqrt{N}\rceil, 5, 30\right), \max(2, N-1)\right).$$

The query prediction is obtained from the fitted label-spreading model's predicted label distribution at the query point. If the labeled context contains only one class, the method predicts that class with probability one.

**RBF label spreading.** The second semi-supervised baseline is `LabelSpreading` with an RBF graph. We use

$$\texttt{kernel} = \texttt{rbf}, \qquad \gamma = 1.0, \qquad \alpha = 0.3, \qquad \texttt{max\_iter} = 3000, \qquad \texttt{tol} = 10^{-4}.$$

The model is fit on the standardized full episode with the query left unlabeled. The predicted label distribution at the query point is used for evaluation. If the labeled context contains only one class, the method predicts that class with probability one.

### B.6.3 VORONOI BASELINES.

For the Voronoi classification task, we use nearest-neighbor baselines rather than the linear and label-spreading baselines above. The baselines do not use the underlying Voronoi sites.

We evaluate 1-NN and 5-NN. For each query feature vector $x_q$, squared Euclidean distances to all context feature vectors are computed:

$$d_i^2 = \|x_i - x_q\|_2^2.$$

The $k$ nearest context points are selected, with $k \in \{1, 5\}$. If $k$ is larger than the number of context points, it is clipped to the context size. The predicted class is obtained by majority vote over the selected neighbors:

$$\hat{y} = \arg \max_{c \in \{1, \dots, K\}} \sum_{i \in \mathcal{N}_k(x_q)} \mathbf{1}\{y_i = c\}.$$

Ties are resolved deterministically by the underlying `argmax`, which selects the lowest-index class.

## C  Analysis of the Mean-Shift Emergent Classifier

We have identified a *family* of attention-driven update rules that take the form:

$$\boldsymbol{X}_{\ell+1} = \boldsymbol{X}_\ell + \alpha'_\ell \boldsymbol{A}_\ell \tilde{\boldsymbol{X}}_\ell,$$
$$\boldsymbol{Y}_{\ell+1} = \boldsymbol{Y}_\ell + \gamma' \boldsymbol{A}_\ell \tilde{\boldsymbol{Y}}_\ell,$$

where $\boldsymbol{X}_0, \boldsymbol{Y}_0$ are defined as in (1) and $\tilde{\boldsymbol{X}}_\ell \in \mathbb{R}^{n \times d}$ and $\tilde{\boldsymbol{Y}}_\ell \in \mathbb{R}^{n \times K}$ are the first $n$ rows of $\boldsymbol{X}_\ell$ and $\boldsymbol{Y}_\ell$, respectively. The matrix $\boldsymbol{A}_t \in \mathbb{R}^{(n+1) \times n}$ is the row-stochastic softmax attention matrix:

$$\boldsymbol{A}_\ell = \sigma(\alpha_\ell \boldsymbol{X}_\ell \tilde{\boldsymbol{X}}_\ell^\top + \gamma_\ell \boldsymbol{Y}_\ell \tilde{\boldsymbol{Y}}_\ell^\top).$$

The transformer instantiates one member of this family during training by selecting the scalars $\{\alpha_\ell, \gamma_\ell, \alpha'_\ell, \gamma'_\ell\}_{\ell \in [L]}$. In the following, we are going to show that this family of updates is a novel approach to classification with surprising behavior and versatility.

Our analysis exposes two key properties of the emergent mean-shift classifier: (1) it reinforces intra-class separation in the training data (Theorem C.1), and (2) it drives the test point towards the correct cluster, assuming it is initialized with some proximity towards that cluster (Theorem C.2)

### C.1  Class and Cone Invariances

For a point $\boldsymbol{x}_i$ let us define as $c_i \in [K]$ the class in which it is originally placed, i.e. $\boldsymbol{y}_i = \boldsymbol{e}_{c_i}$. For each class $c \in [K]$, we define its population as $S_c := \{i : \boldsymbol{y}_i = \boldsymbol{e}_c\}$.

First, we focus on the evolution of the labels $\boldsymbol{Y}_t$. We show that in the regime $\gamma \gg \alpha$ the labels of the training points all remain pointing to their original class.

**Lemma 1.** *Let $\kappa = \gamma/\alpha$. For all $t \geq 0$, the attention weight on any incorrect class index $k \notin S_{c(j)}$ decays exponentially with $\kappa$:*

$$A_{jk}^{(t)} \leq e^{-\alpha \kappa \eta} \cdot \frac{e^{\alpha \langle x_j^{(t)}, x_k^{(t)} \rangle}}{Z_{j,in}^{(t)}},$$

*where $Z_{j,in}$ is the partition function restricted to the correct class. Consequently, as $\kappa \to \infty$, the 'leakage' of attention weights across class boundaries vanishes and the dynamics recover the exact label evolution $y_j = (1 + \gamma')^t \cdot \boldsymbol{e}_{c(j)}$.*

*Proof.* We first analyze the structure of the attention matrix $A^{(t)}$ in terms of the ratio $\kappa = \gamma/\alpha$.

Let $S_{c(j)} = \{k \in [n] : c(k) = c(j)\}$ be the set of indices in the same class as $j$. We assume a label separation margin $\langle y_j, y_k \rangle = \eta > 0$ for $k \in S_{c(j)}$ and 0 otherwise. Substituting $\gamma = \kappa\alpha$, the partition function is:

$$Z_j = e^{\alpha\kappa\eta} \sum_{s \in S_{c(j)}} e^{\alpha\langle x_j, x_s \rangle} + \sum_{s \notin S_{c(j)}} e^{\alpha\langle x_j, x_s \rangle}.$$

We analyze the probability mass $A_{jk}$ in two cases:

**Case 1: Different Class ($k \notin S_{c(j)}$).** The attention weight is strictly bounded by the label margin $\kappa\eta$. Dividing the numerator and denominator by $e^{\alpha\kappa\eta}$:

$$A_{jk} = \frac{e^{-\alpha\kappa\eta}e^{\alpha\langle x_j, x_k \rangle}}{\sum_{s \in S_{c(j)}} e^{\alpha\langle x_j, x_s \rangle} + e^{-\alpha\kappa\eta}\sum_{s \notin S_{c(j)}} e^{\alpha\langle x_j, x_s \rangle}} \leq e^{-\alpha\kappa\eta} \cdot \frac{e^{\alpha\langle x_j, x_k \rangle}}{\sum_{s \in S_{c(j)}} e^{\alpha\langle x_j, x_s \rangle}}.$$

Thus, the leakage to incorrect classes decays exponentially with $\kappa$.

**Case 2: Same Class ($k \in S_{c(j)}$).** Factorizing $e^{\alpha\kappa\eta}$ from the numerator and denominator yields the exact expression:

$$A_{jk} = \frac{e^{\alpha\langle x_j, x_k \rangle}}{\sum_{s \in S_{c(j)}} e^{\alpha\langle x_j, x_s \rangle} + e^{-\alpha\kappa\eta}\sum_{s \notin S_{c(j)}} e^{\alpha\langle x_j, x_s \rangle}}.$$

Taking the limit $\kappa \to \infty$ eliminates the cross-class term in the denominator, recovering the softmax over the class cluster:

$$\lim_{\kappa \to \infty} A_{jk} = \frac{e^{\alpha\langle x_j, x_k \rangle}}{\sum_{s \in S_{c(j)}} e^{\alpha\langle x_j, x_s \rangle}}.$$

If we further assumed the feature scale $\alpha \to 0$ (relative to the label signal), the exponential terms approach unity, recovering a uniform distribution over the class-cluster:

$$\lim_{\alpha \to 0} \left( \lim_{\kappa \to \infty} A_{jk} \right) = \frac{1}{|S_{c(j)}|}.$$

Of course, in this limit, the test point observes no nontrivial evolution. Henceforth, we will exploit the block-diagonal structure of the training attention with (implicitly) nonzero $\alpha$.

**Label Evolution.** We now proceed by induction on the labels using the asymptotic limit derived above ($A_{jk} = \frac{1}{|S_{c(j)}|}\mathbb{I}[c(k) = c(j)]$). For $t = 0$, the statement holds by definition. Assuming the statement is true for some $t > 0$, the labels evolve as:

$$y_j^{(t+1)} = y_j^{(t)} + \gamma' \sum_{k=1}^n A_{jk}^{(t)} y_k^{(t)}$$
$$= y_j^{(t)} + \gamma' \sum_{k:c(k)=c(j)} A_{jk}^{(t)} y_k^{(t)}.$$

By the inductive hypothesis, for all $k \in S_{c(j)}$, $y_k^{(t)} = y_j^{(t)}$. Since $A_{jk}^{(t)}$ is a probability distribution concentrated on $S_{c(j)}$, the sum is equal to $y_j^{(t)}$. Thus,

$$y_j^{(t+1)} = y_j^{(t)} + \gamma' y_j^{(t)} = (1 + \gamma')y_j^{(t)}.$$

This completes the proof. $\square$

Next, we show that class membership is preserved geometrically as well. We define the polyhedral cone $C_c$ capturing all the points classified as class $c$ with positive margin against all other classes.

$$\mathcal{C}_c := \{x \in \mathbb{R}^d : \langle w_c - w_{c'}, x \rangle > 0 \;\; \forall c' \neq c\}.$$

We show that in the limit case $\gamma \gg \alpha$ each point $x_i^{(t)}$ throughout time remains in its original cone. We show this by induction on the timestep $t$.

**Lemma 2** (Invariant Cones). *Assume that at time $t$,*

$$\boldsymbol{x}_i^{(t)} \in \mathcal{C}_{y_i} \quad \forall i.$$

*In the regime $\kappa \to \infty$, the same condition holds at time $t+1$.*

*Proof.* By Lemma 1, in the high-$\gamma$ regime, the attention matrix becomes block-diagonal with respect to the class partition:

$$\mathbf{A}_{\text{train}}^{(t)} = \text{diag}\big(\mathbf{A}_{(1)}^{(t)}, \dots, \mathbf{A}_{(K)}^{(t)}\big),$$

where each block $\mathbf{A}_{(c)}^{(t)}$ is row-stochastic and has nonnegative entries.

Fix a class $c$ and a point $i \in S_c$. The feature update satisfies

$$\boldsymbol{x}_i^{(t+1)} = \boldsymbol{x}_i^{(t)} + \alpha' \sum_{j \in S_c} A_{ij}^{(t)} \boldsymbol{x}_j^{(t)}.$$

Let $c' \neq c$. Taking inner products with $\boldsymbol{w}_c - \boldsymbol{w}_{c'}$ gives

$$\begin{aligned}
\langle \boldsymbol{w}_c - \boldsymbol{w}_{c'}, \boldsymbol{x}_i^{(t+1)} \rangle &= \langle \boldsymbol{w}_c - \boldsymbol{w}_{c'}, \boldsymbol{x}_i^{(t)} \rangle \\
&\quad + \alpha' \sum_{j \in S_c} A_{ij}^{(t)} \langle \boldsymbol{w}_c - \boldsymbol{w}_{c'}, \boldsymbol{x}_j^{(t)} \rangle.
\end{aligned}$$

All terms on the right-hand side are strictly positive by the induction hypothesis and nonnegativity of the weights. Thus $\boldsymbol{x}_i^{(t+1)} \in \mathcal{C}_c$. $\square$

## C.2 Monotonic Global Directional Margin

Consider the **centroid** of a class $c \in [K]$ defined as

$$\boldsymbol{\mu}_c := \frac{1}{|S_c|} \sum_{i \in S_c} \boldsymbol{x}_i.$$

To quantify separation in the multiclass setting, we define a global directional margin. Intuitively, it is not enough to show that $\|\boldsymbol{\mu}_c^{(t)} - \boldsymbol{\mu}_{c'}^{(t)}\|_2$ grows to establish class separation because then the separation could be on a direction orthogonal to the boundary $\boldsymbol{w}_c - \boldsymbol{w}_{c'}$. Hence we define

$$\mathcal{M}_t := \sum_{c=1}^{K} \sum_{\substack{c'=1 \\ c' \neq c}}^{K} \langle \boldsymbol{w}_c - \boldsymbol{w}_{c'}, \boldsymbol{\mu}_c^{(t)} - \boldsymbol{\mu}_{c'}^{(t)} \rangle.$$

We show that this margin increases in $t$:

**Theorem C.1.** *In the limit $\kappa \to \infty$, the global directional margin $\mathcal{M}_t$ is strictly increasing in $t$.*

*Proof.* We analyze the evolution of a single centroid. Since $\kappa \to \infty$, recall by Lemma 1 that in every timestep the class of each token remains the same. Fix a class $c$. Using the block-diagonal structure of $\mathbf{A}_{\text{train}}^{(t)}$, the centroid update is

$$\begin{aligned}
\boldsymbol{\mu}_c^{(t+1)} &= \frac{1}{|S_c|} \sum_{i \in S_c} \boldsymbol{x}_i^{(t+1)} \\
&= \frac{1}{|S_c|} \sum_{i \in S_c} \left( \boldsymbol{x}_i^{(t)} + \alpha' \sum_{j \in S_c} A_{ij}^{(t)} \boldsymbol{x}_j^{(t)} \right) &\left( x_i^{(t+1)} = \sum_{j \in S_c} A_{ij}^{(t)} x_j^{(t)} \text{ when } \gamma \gg \alpha \right) \\
&= \boldsymbol{\mu}_c^{(t)} + \alpha' \sum_{j \in S_c} \left( \frac{1}{|S_c|} \sum_{i \in S_c} A_{ij}^{(t)} \right) \boldsymbol{x}_j^{(t)} \\
&= \boldsymbol{\mu}_c^{(t)} + \alpha' \sum_{j \in S_c} \pi_j^{(t)} \boldsymbol{x}_j^{(t)},
\end{aligned}$$

where

$$\pi_j^{(t)} := \frac{1}{|S_c|} \sum_{i \in S_c} A_{ij}^{(t)}$$

defines a probability distribution over $S_c$.

Fix $c' \neq c$. Taking inner products with $\boldsymbol{w}_c - \boldsymbol{w}_{c'}$ yields

$$\langle \boldsymbol{w}_c - \boldsymbol{w}_{c'}, \boldsymbol{\mu}_c^{(t+1)} \rangle = \langle \boldsymbol{w}_c - \boldsymbol{w}_{c'}, \boldsymbol{\mu}_c^{(t)} \rangle$$
$$+ \alpha' \sum_{j \in S_c} \pi_j^{(t)} \langle \boldsymbol{w}_c - \boldsymbol{w}_{c'}, \boldsymbol{x}_j^{(t)} \rangle.$$

By the invariant cone lemma, each inner product in the sum is strictly positive, implying

$$\langle \boldsymbol{w}_c - \boldsymbol{w}_{c'}, \boldsymbol{\mu}_c^{(t+1)} \rangle > \langle \boldsymbol{w}_c - \boldsymbol{w}_{c'}, \boldsymbol{\mu}_c^{(t)} \rangle.$$

Summing this inequality over all $c' \neq c$ and then over all classes $c$ shows that $\mathcal{M}_{t+1} > \mathcal{M}_t$. $\qquad\square$

## C.3 Test Point Dynamics

We now proceed to our main theoretical result - the analysis of the test point dynamics. In the high $\gamma$ regime ($\gamma \gg \alpha$), the initial test label $y^{(0)}$ is uniform across classes. Consequently, the label-driven component of the attention mechanism is initially uninformative, and the test point's trajectory is governed primarily by the feature dot products $\langle x^{(t)}, x_i^{(t)} \rangle$.

This observation motivates our analysis strategy: we approximate the training data dynamics $(X^{(t)}, Y^{(t)})$ using the simplified label-driven limit (where points cluster perfectly by class), while retaining the full, coupled dynamics for the test point $x_{test}$. This approximation captures the two-phase nature of the inference process. In the early phase, feature similarity breaks the symmetry, guiding the test point toward the correct cluster and reinforcing the correct class label. Once the label margin $\Delta_y^{(t)}$ becomes sufficiently large, the dynamics transition to a label-driven regime, resulting in the asymptotic geometric convergence of the test point and confidence.

The following equations describe the coupled trajectories of the test point $x_{\text{test}}$ and its label embedding.

$$x^{(t+1)} = x^{(t)} + \alpha' \sum_{i=1}^{n} A_i^{(t)} x_i^{(t)} \tag{15}$$

$$y^{(t+1)} = y^{(t)} + \alpha' \sum_{i=1}^{n} A_i^{(t)} y_i^{(t)} \tag{16}$$

where $A_i^{(t)}$ denote the $i$th coordinate of the test-token attention, i.e., $A_i^{(t)} = (A_{\text{test}}^{(t)})_i$, with

$$A_{\text{test}}^{(t)} = \sigma(\alpha x^{(t)} X_t^\top + \gamma y^{(t)} Y_t^\top).$$

We define the class-level attention masses

$$p_c^{(t)} := \sum_{j \in S_c} A_j^{(t)},$$

and the class logits

$$s_c^{(t)} := \log Z_c^{(t)} + \gamma \lambda_t y_{t,c}, \quad Z_c^{(t)} := \sum_{j \in S_c} \exp\big( \alpha \langle x^{(t)}, x_j^{(t)} \rangle \big),$$

where $\lambda_t = (1 + \alpha')^t$ is the norm of the training label vectors after $t$ steps. By observing that under a class-wise block diagonal attention dynamics, $\forall t, Y_i^t = \lambda_t \mathbf{e}_{c(i)}$, we find that

$$A_j^{(t)} \propto \exp(\alpha \langle x^{(t)}, x_j^t \rangle + \gamma \langle y_t, Y_j \rangle) = \exp(\alpha \langle x^{(t)}, x_j^t \rangle + \gamma \lambda_t y_{t,c}),$$

we find that

$$\sum_{j \in S_c} A_j^{(t)} \propto \sum_{j \in S_c} e^{\gamma y_{t,c}} e^{\alpha \langle x^{(t)}, x_j^{(t)} \rangle} = \exp(s_c^{(t)}).$$

Since $\sum_j A_j^{(t)} = \sum_c \sum_{j \in S_c} A_j^{(t)}$, it follows thus that

$$p_c^{(t)} = \frac{e^{s_c^{(t)}}}{\sum_r e^{s_r^{(t)}}}.$$

Suppose the test point $x_{\text{test}}$ belongs to class $c^\star$. We begin by establishing structural bounds on the geometric evolution.

**Lemma 3.** *Assume the following initialization at time $t = 0$:*

*(i) (Test margin) Define*

$$R_0 := \min_{j \in S_{c^\star}} \langle x^{(0)}, x_j^{(0)} \rangle, \qquad L_0 := \max_{j \notin S_{c^\star}} \langle x^{(0)}, x_j^{(0)} \rangle,$$

*and assume $\Delta_0 := R_0 - L_0 > 0$.*

*(ii) (Training separation) Define*

$$\rho_0 := \min_{i,j \in S_{c^\star}} \langle x_i^{(0)}, x_j^{(0)} \rangle, \qquad \Lambda_0 := \max_{i \in S_{c^\star}, j \notin S_{c^\star}} \langle x_i^{(0)}, x_j^{(0)} \rangle,$$

*and assume $\Gamma_0 := \rho_0 - \Lambda_0 > 0$.*

*(iii) (Norm bound) $\|x^{(0)}\|_2 \le 1$ and $\|x_i^{(0)}\|_2 \le 1$ for all $i$.*

*For each $t \ge 0$, define the test-point quantities*

$$R_t := \min_{j \in S_{c^\star}} \langle x^{(t)}, x_j^{(t)} \rangle, \qquad L_t := \max_{j \notin S_{c^\star}} \langle x^{(t)}, x_j^{(t)} \rangle, \qquad \Delta_t := R_t - L_t,$$

*and the training-point quantities*

$$\rho_t := \min_{i,j \in S_{c^\star}} \langle x_i^{(t)}, x_j^{(t)} \rangle, \qquad \Lambda_t := \max_{i \in S_{c^\star}, j \notin S_{c^\star}} \langle x_i^{(t)}, x_j^{(t)} \rangle, \qquad \Gamma_t := \rho_t - \Lambda_t.$$

*Finally, let $\widetilde{\Delta}_t := \min(\Delta_t, \Gamma_t)$. Then, the following hold for all $t \ge 0$:*

*(a) (Norm growth) For all $i$, $\max_{j \in S_{c(i)}} \|x_j^{(t)}\|_2 \le (1 + \alpha')^t$.*

*(b) (Training margin recursion)*

$$\Gamma_{t+1} \ge (1 + \alpha')^2 \Gamma_t.$$

*(c) (Test margin recursion)*

$$\Delta_{t+1} \ge (1 + \alpha')\Delta_t + \alpha'(1 + \alpha')p_{c^\star}^{(t)}\Gamma_t - 2\alpha'(1 - p_{c^\star}^{(t)})(1 + \alpha')^{2t+1}.$$

*(d) (Effective margin recursion) The quantity $\widetilde{\Delta}_t$ satisfies*

$$\widetilde{\Delta}_{t+1} \ge \left(1 + \alpha' + \alpha'(1 + \alpha')p_{c^\star}^{(t)}\right)\widetilde{\Delta}_t - 2\alpha'(1 - p_{c^\star}^{(t)})(1 + \alpha')^{2t+1}. \tag{17}$$

*Proof.* **Part (a):** Recall the update rule for a training point $x_j$ belonging to class $c(j)$:

$$x_j^{(t+1)} = x_j^{(t)} + \alpha' \sum_{k \in S_{c(j)}} A_{jk}^{(t)} x_k^{(t)}.$$

Taking the Euclidean norm of both sides and applying the triangle inequality yields:

$$\|x_j^{(t+1)}\|_2 = \left\| x_j^{(t)} + \alpha' \sum_{k \in S_{c(j)}} A_{jk}^{(t)} x_k^{(t)} \right\|_2 \le \|x_j^{(t)}\|_2 + \alpha' \sum_{k \in S_{c(j)}} A_{jk}^{(t)} \|x_k^{(t)}\|_2.$$

Let $M_t := \max_{k \in S_{c(j)}} \|x_k^{(t)}\|_2$ denote the maximum feature norm within the class at iteration $t$. We bound:

$$\sum_{k \in S_{c(j)}} A_{jk}^{(t)} \|x_k^{(t)}\|_2 \leq \sum_{k \in S_{c(j)}} A_{jk}^{(t)} M_t = M_t \left( \sum_{k \in S_{c(j)}} A_{jk}^{(t)} \right).$$

By row stochasticity, we have:

$$\|x_j^{(t+1)}\|_2 \leq \|x_j^{(t)}\|_2 + \alpha' M_t \leq (1 + \alpha') M_t$$

Taking the maximum over all $j \in S_{c(j)}$ on the left-hand side establishes part (a).

**Part (b):** For $i, j \in S_{c^\star}$, write

$$\Delta x_i^{(t)} := \alpha' \sum_{k \in S_{c^\star}} A_{ik}^{(t)} x_k^{(t)}, \qquad \Delta x_j^{(t)} := \alpha' \sum_{k \in S_{c^\star}} A_{jk}^{(t)} x_k^{(t)}.$$

Since each sum is a convex combination of points in $S_{c^\star}$, every inner product of the form $\langle \Delta x_i^{(t)}/\alpha', x_j^{(t)} \rangle$, $\langle x_i^{(t)}, \Delta x_j^{(t)}/\alpha' \rangle$, and $\langle \Delta x_i^{(t)}/\alpha', \Delta x_j^{(t)}/\alpha' \rangle$ is bounded below by $\rho_t$. Hence

$$\langle x_i^{(t+1)}, x_j^{(t+1)} \rangle \geq \langle x_i^{(t)}, x_j^{(t)} \rangle + 2\alpha' \rho_t + \alpha'^2 \rho_t \geq (1 + \alpha')^2 \rho_t.$$

Taking the minimum over $i, j \in S_{c^\star}$ gives

$$\rho_{t+1} \geq (1 + \alpha')^2 \rho_t.$$

Now let $i \in S_{c^\star}$ and $j \notin S_{c^\star}$. Since $\Delta x_i^{(t)}/\alpha'$ is a convex combination of points in $S_{c^\star}$ while $\Delta x_j^{(t)}/\alpha'$ is a convex combination of points in the incorrect class $S_{c(j)}$, every inner product of the form $\langle \Delta x_i^{(t)}/\alpha', x_j^{(t)} \rangle$, $\langle x_i^{(t)}, \Delta x_j^{(t)}/\alpha' \rangle$, and $\langle \Delta x_i^{(t)}/\alpha', \Delta x_j^{(t)}/\alpha' \rangle$ is bounded above by $\Lambda_t$. Therefore

$$\langle x_i^{(t+1)}, x_j^{(t+1)} \rangle \leq \langle x_i^{(t)}, x_j^{(t)} \rangle + 2\alpha' \Lambda_t + \alpha'^2 \Lambda_t \leq (1 + \alpha')^2 \Lambda_t.$$

Taking the maximum over $i \in S_{c^\star}$ and $j \notin S_{c^\star}$ gives

$$\Lambda_{t+1} \leq (1 + \alpha')^2 \Lambda_t.$$

Subtracting yields

$$\Gamma_{t+1} = \rho_{t+1} - \Lambda_{t+1} \geq (1 + \alpha')^2 (\rho_t - \Lambda_t) = (1 + \alpha')^2 \Gamma_t.$$

**Part (c):** Let $\Delta x^{(t)} = \alpha' \sum_i A_i^{(t)} x_i^{(t)}$ denote the update to the test point, and $\Delta x_j^{(t)} = \alpha' \sum_{k \in S_{c(j)}} A_{jk}^{(t)} x_k^{(t)}$ denote the update to the training point $j$. Expanding the inner product by linearity:

$$\langle x^{(t+1)}, x_j^{(t+1)} \rangle = \langle x^{(t)} + \Delta x^{(t)}, \ x_j^{(t)} + \Delta x_j^{(t)} \rangle$$
$$= \langle x^{(t)}, x_j^{(t)} \rangle + \underbrace{\langle \Delta x^{(t)}, x_j^{(t)} \rangle}_{\text{Term 1 (Test Update } U_j)} + \underbrace{\langle x^{(t)}, \Delta x_j^{(t)} \rangle}_{\text{Term 2 (Training Update } V_j)} + \underbrace{\langle \Delta x^{(t)}, \Delta x_j^{(t)} \rangle}_{\text{Quadratic term}}.$$

We first explicitly **bound the quadratic cross-term** resulting from the coupled updates. Let $Q_j^{(t)} := \langle \Delta x^{(t)}, \Delta x_j^{(t)} \rangle$. Recall that the training point update is $\Delta x_j^{(t)} = \alpha' \sum_{k \in S_{c(j)}} A_{jk}^{(t)} x_k^{(t)}$. Since $A_{jk}^{(t)}$ is row-stochastic, $\Delta x_j^{(t)}/\alpha'$ represents a convex combination of feature vectors within the cluster $c(j)$. We define this cluster-average vector as $\nu_{c(j)}^{(t)} := \sum_{k \in S_{c(j)}} A_{jk}^{(t)} x_k^{(t)}$. The quadratic term can thus be written as:

$$Q_j^{(t)} = \langle \alpha' \sum_{i=1}^{n} A_i^{(t)} x_i^{(t)}, \alpha' \nu_{c(j)}^{(t)} \rangle = \alpha'^2 \sum_{i=1}^{n} A_i^{(t)} \langle x_i^{(t)}, \nu_{c(j)}^{(t)} \rangle.$$

We derive bounds for this term depending on whether $j$ belongs to the correct class $c^*$.

**Lower Bound** ($j \in S_{c^\star}$): If $j$ is in the correct class, then $\nu_{c(j)}^{(t)}$ is a convex combination of points in $S_{c^\star}$. We split the summation over the test attention masses into the correct class ($i \in S_{c^\star}$) and incorrect classes ($i \notin S_{c^\star}$).

- For $i \in S_{c^*}$, the inner product $\langle x_i^{(t)}, \nu_{c(j)}^{(t)} \rangle$ is a weighted average of intra-class inner products. By the definition of $\rho_t$, $\langle x_i^{(t)}, x_k^{(t)} \rangle \geq \rho_t$ for all $k \in S_{c^*}$. Thus, $\langle x_i^{(t)}, \nu_{c(j)}^{(t)} \rangle \geq \rho_t$.

- For $i \notin S_{c^*}$, we apply the global norm bound. Since $\|x\| \leq (1 + \alpha')^t$, the inner product is lower-bounded by $-(1 + \alpha')^{2t}$.

Combining these, and noting that $\sum_{i \in S_{c^*}} A_i^{(t)} = p_{c^*}^{(t)}$:

$$Q_j^{(t)} \geq \alpha'^2 \left( p_{c^*}^{(t)} \rho_t - (1 - p_{c^*}^{(t)})(1 + \alpha')^{2t} \right).$$

**Upper Bound ($j \notin S_{c^*}$):** If $j$ is in an incorrect class $c \neq c^*$, then $\nu_{c(j)}^{(t)}$ consists of points from $S_c$.

- For $i \in S_{c^*}$, the inner product $\langle x_i^{(t)}, \nu_{c(j)}^{(t)} \rangle$ represents cross-class similarity between $c^*$ and $c$. By the definition of $\Lambda_t$, $\langle x_i^{(t)}, x_k^{(t)} \rangle \leq \Lambda_t$ for every $k \in S_{c(j)}$. Thus, $\langle x_i^{(t)}, \nu_{c(j)}^{(t)} \rangle \leq \Lambda_t$.

- For $i \notin S_{c^*}$, we again apply the worst-case norm bound $(1 + \alpha')^{2t}$.

This yields the upper bound:

$$Q_j^{(t)} \leq \alpha'^2 \left( p_{c^*}^{(t)} \Lambda_t + (1 - p_{c^*}^{(t)})(1 + \alpha')^{2t} \right).$$

We explicitly bound the contributions from the test update $U_j$ and the training update $V_j$. We split the summations into indices belonging to the correct class $S_{c^\star}$ and the incorrect classes $S_{c^\star}^c$.

**1. Lower Bound for $R_{t+1}$ (Case $j \in S_{c^\star}$).** For a point $j$ in the correct class, we want to lower bound both update terms.

- **Test Update ($U_j$):**
$$U_j = \alpha' \sum_{i \in S_{c^\star}} A_i^{(t)} \langle x_i^{(t)}, x_j^{(t)} \rangle + \alpha' \sum_{i \notin S_{c^\star}} A_i^{(t)} \langle x_i^{(t)}, x_j^{(t)} \rangle.$$

  For $i \in S_{c^\star}$ we have $\langle x_i, x_j \rangle \geq \rho_t$, while for $i \notin S_{c^\star}$ by Cauchy-Schwarz and the norm bounds from Part (a) we get a lower bound of $-(1 + \alpha')^{2t}$, so
$$U_j \geq \alpha' \left( p_{c^\star}^{(t)} \rho_t - (1 - p_{c^\star}^{(t)})(1 + \alpha')^{2t} \right).$$

- **Training Update ($V_j$):**
$$V_j = \alpha' \sum_{k \in S_{c(j)}} A_{jk}^{(t)} \langle x^{(t)}, x_k^{(t)} \rangle.$$

  Since $j \in S_{c^\star}$, the sum runs over $k \in S_{c^\star}$. By definition, $\langle x^{(t)}, x_k^{(t)} \rangle \geq R_t$. Thus:
$$V_j \geq \alpha' \sum_{k \in S_{c^\star}} A_{jk}^{(t)} R_t = \alpha' R_t.$$

Combining these with the quadratic lower bound gives the total lower bound:

$$R_{t+1} \geq R_t + \alpha' R_t + \alpha'(1 + \alpha') \left( p_{c^\star}^{(t)} \rho_t - (1 - p_{c^\star}^{(t)})(1 + \alpha')^{2t} \right)$$

**2. Upper Bound for $L_{t+1}$ (Case $j \notin S_{c^\star}$).** For a point $j$ in an incorrect class, we want to upper bound both update terms.

- **Test Update ($U_j$):**
$$U_j = \alpha' \sum_{i \in S_{c^\star}} A_i^{(t)} \langle x_i^{(t)}, x_j^{(t)} \rangle + \alpha' \sum_{i \notin S_{c^\star}} A_i^{(t)} \langle x_i^{(t)}, x_j^{(t)} \rangle.$$

  Using $\langle x_i, x_j \rangle \leq \Lambda_t$ for cross-class pairs and the Cauchy-Schwarz norm bound $(1 + \alpha')^{2t}$ otherwise:
$$U_j \leq \alpha' \left( p_{c^\star}^{(t)} \Lambda_t + (1 - p_{c^\star}^{(t)})(1 + \alpha')^{2t} \right).$$

- **Training Update ($V_j$):**

$$V_j = \alpha' \sum_{k \in S_{c(j)}} A_{jk}^{(t)} \langle x^{(t)}, x_k^{(t)} \rangle.$$

Since $j \notin S_{c^\star}$, the class $c(j) \neq c^\star$. Thus, any $k \in S_{c(j)}$ is not in the correct class. By definition, $\langle x^{(t)}, x_k^{(t)} \rangle \leq L_t$. Thus:

$$V_j \leq \alpha' \sum_{k \in S_{c(j)}} A_{jk}^{(t)} L_t = \alpha' L_t.$$

Combining these with the quadratic upper bound gives the total upper bound:

$$L_{t+1} \leq L_t + \alpha' L_t + \alpha'(1+\alpha') \left( p_{c^\star}^{(t)} \Lambda_t + (1 - p_{c^\star}^{(t)})(1+\alpha')^{2t} \right)$$

Subtracting upper from lower bound yields

$$\Delta_{t+1} \geq (1+\alpha')\Delta_t + \alpha'(1+\alpha')p_{c^\star}^{(t)}(\rho_t - \Lambda_t) - 2\alpha'(1 - p_{c^\star}^{(t)})(1+\alpha')^{2t+1},$$

which is exactly the claimed recursion for $\Delta_t$.

**Part (d):** Since $p_{c^\star}^{(t)} \leq 1$, part (b) implies

$$\Gamma_{t+1} \geq (1+\alpha')^2 \Gamma_t \geq \left(1 + \alpha' + \alpha'(1+\alpha')p_{c^\star}^{(t)}\right) \Gamma_t.$$

Using $\widetilde{\Delta}_t \leq \Delta_t$ and $\widetilde{\Delta}_t \leq \Gamma_t$, part (c) gives

$$\Delta_{t+1} \geq \left(1 + \alpha' + \alpha'(1+\alpha')p_{c^\star}^{(t)}\right) \widetilde{\Delta}_t - 2\alpha'(1 - p_{c^\star}^{(t)})(1+\alpha')^{2t+1}.$$

Combining the last two displays and taking the minimum proves the recursion for $\widetilde{\Delta}_{t+1}$. $\qquad \square$

We finally prove that the margin $\Delta_t$ grows indefinitely and that the label prediction remains correct, meaning that the test point is attracted to the correct class both geometrically and with its label. This requires a coupled induction on the geometric margin $\Delta_t$ and the label margin $\Delta_y^{(t)} := y_{c^\star}^{(t)} - y_c^{(t)}$ for any class $c \neq c^\star$.

**Theorem C.2.** *Assume $\alpha' < 1$ and the initialization satisfies:*

$$\widetilde{\Delta}_0 > 0, \quad \alpha \Delta_0 > \log(K) + \log(1 + 2/\widetilde{\Delta}_0) \quad \text{and} \quad \Delta_y^{(0)} \geq 0 \quad (\forall c \neq c^\star). \tag{18}$$

*Then, for all $t \geq 0$:*

1. *The geometric margin grows geometrically: $\Delta_t \geq \Delta_0(1+\alpha')^t$.*

2. *The label margin is non-decreasing: $\Delta_y^{(t)} \geq 0$.*

*Proof.* We proceed by induction on $t$. The base case $t = 0$ holds by assumption. Assume the hypotheses hold for step $t$.

**Bounding the Logit Gap:** Consider the difference in logits between the correct class $c^\star$ and any incorrect class $c$:

$$s_{c^\star}^{(t)} - s_c^{(t)} = \underbrace{\log \frac{Z_{c^\star}^{(t)}}{Z_c^{(t)}}}_{\text{Geometric Term}} + \gamma \underbrace{(y_{c^\star}^{(t)} - y_c^{(t)})}_{\text{Label Term } \Delta_y^{(t)}}.$$

By the definition of $R_t$ and $L_t$, we have $Z_{c^\star}^{(t)} \geq |S_{c^\star}|e^{\alpha R_t}$ and $Z_c^{(t)} \leq |S_c|e^{\alpha L_t}$. Assuming balanced classes ($|S_{c^\star}| = |S_c|$):

$$\log \frac{Z_{c^\star}^{(t)}}{Z_c^{(t)}} \geq \alpha(R_t - L_t) = \alpha \Delta_t.$$

Using the inductive hypothesis, $\forall c \neq c^*, y_{t,c^*} - y_{t,c} \geq \Delta_y^{(t)} \geq 0$. Thus:

$$s_{c^\star}^{(t)} - s_c^{(t)} = \log \frac{Z_{c^*}^{(t)}}{Z_c^{(t)}} + \gamma(y_{t,c^*} - y_{t,c}) \geq \alpha\Delta_t. \tag{19}$$

**Bounding Attention Leakage:** The probability mass on incorrect classes is $1 - p_{c^\star}^{(t)} = \sum_{c \neq c^*} p_c^{(t)}$. Using Eq. 19:

$$1 - p_{c^\star}^{(t)} = \sum_{c \neq c^*} \frac{e^{s_c^{(t)}}}{\sum_r e^{s_r^{(t)}}} \leq \sum_{c \neq c^\star} e^{-(s_{c^\star}^{(t)} - s_c^{(t)})} \leq \sum_{c \neq c^\star} e^{-\alpha\Delta_t} \leq Ke^{-\alpha\Delta_t}.$$

Using the inductive hypothesis $\Delta_t \geq \Delta_0(1 + \alpha')^t$, this leakage term decays super-exponentially.

**Inductive step on $\Delta_{t+1}$** From Lemma 3(b), the training margin obeys

$$\Gamma_t \geq \Gamma_0(1 + \alpha')^{2t}.$$

Since $\widetilde{\Delta}_0 = \min(\Delta_0, \Gamma_0)$, we in particular have $\Gamma_0 \geq \widetilde{\Delta}_0$.

From Lemma 3(c), the test margin updates as

$$\Delta_{t+1} \geq (1 + \alpha')\Delta_t + \alpha'(1 + \alpha')p_{c^\star}^{(t)}\Gamma_t - 2\alpha'(1 - p_{c^\star}^{(t)})(1 + \alpha')^{2t+1}.$$

Using the lower bound on $\Gamma_t$ gives

$$\Delta_{t+1} \geq (1 + \alpha')\Delta_t + \alpha'(1 + \alpha')(1 + \alpha')^{2t}\left(p_{c^\star}^{(t)}\Gamma_0 - 2(1 - p_{c^\star}^{(t)})\right).$$

Therefore it suffices to show

$$p_{c^\star}^{(t)}\Gamma_0 \geq 2(1 - p_{c^\star}^{(t)}).$$

This inequality is equivalent to

$$(1 - p_{c^\star}^{(t)})(\Gamma_0 + 2) \leq \Gamma_0.$$

By the leakage estimate above and the inductive lower bound on $\Delta_t$,

$$(1 - p_{c^\star}^{(t)})(\Gamma_0 + 2) \leq Ke^{-\alpha\Delta_t}(\Gamma_0 + 2) \leq Ke^{-\alpha\Delta_0}(\Gamma_0 + 2).$$

Since $\Gamma_0 \geq \widetilde{\Delta}_0$, we have

$$Ke^{-\alpha\Delta_0}(\Gamma_0 + 2) \leq Ke^{-\alpha\Delta_0}\Gamma_0\left(1 + \frac{2}{\widetilde{\Delta}_0}\right).$$

$$Ke^{-\alpha\Delta_0}\left(1 + \frac{2}{\widetilde{\Delta}_0}\right) \leq 1$$

by Eq. (18), and hence

$$(1 - p_{c^\star}^{(t)})(\Gamma_0 + 2) \leq \Gamma_0.$$

Therefore $\Delta_{t+1} \geq (1 + \alpha')\Delta_t$, which closes the induction.

**Label Margin Induction ($\Delta_y^{(t+1)}$).** Finally, recall the update rule for the test label embedding: $y^{(t+1)} = y^{(t)} + \alpha'\sum_{i=1}^n A_i^{(t)}y_i^{(t)}$. We track the margin between the correct class component $c^\star$ and an incorrect class component $c$. From Lemma 1, the training labels maintain their class-alignment throughout the dynamics, satisfying $y_i^{(t)} = \lambda_t e_{c(i)}$ for some scalar scaling factor $\lambda_t > 0$. Substituting this into the update rule, the $k$-th component evolves as:

$$y_k^{(t+1)} = y_k^{(t)} + \alpha'\lambda_t \sum_{i=1}^n A_i^{(t)}\mathbb{I}(c(i) = k) = y_k^{(t)} + \alpha'\lambda_t p_k^{(t)}.$$

Now, considering the margin $\Delta_y^{(t)} = y_{c^\star}^{(t)} - y_c^{(t)}$:

$$\Delta_y^{(t+1)} = (y_{c^\star}^{(t)} + \alpha'\lambda_t p_{c^\star}^{(t)}) - (y_c^{(t)} + \alpha'\lambda_t p_c^{(t)}) = \Delta_y^{(t)} + \alpha'\lambda_t \underbrace{(p_{c^\star}^{(t)} - p_c^{(t)})}_{\text{Drift Term}}.$$

In Step 1, we established that the logit gap $s_{c^\star}^{(t)} - s_c^{(t)} \geq \alpha\Delta_t$ is strictly positive. Since the softmax function is monotonic, a larger logit implies a larger probability mass: $s_{c^\star}^{(t)} > s_c^{(t)} \implies p_{c^\star}^{(t)} > p_c^{(t)}$. Therefore, the drift term is strictly positive. Combined with the inductive hypothesis $\Delta_y^{(t)} \geq 0$, it follows that the label margin strictly increases: $\Delta_y^{(t+1)} > \Delta_y^{(t)} \geq 0$.

This completes the induction.

**Growth of the Label Margin.** Coupling the above with the growing size of $\lambda_t$ in turn induces a growth in the label margin. Indeed, let

$$t_0 := \inf\{t : 1 - 2Ke^{-\alpha\Delta_t} \geq 1/2\}.$$

Then note that for all $t \geq t_0$,

$$p_{c^*}^{(t)} - p_c^{(t)} \geq p_{c^*}^{(t)} - (1 - p_{c^*}^{(t)}) = 1 - 2(1 - p_{c^*}^{(t)}) \geq 1 - 2Ke^{-\alpha\Delta_t} \geq \frac{1}{2}.$$

Thus, for all $t \geq t_0$,

$$\Delta_y^{(t+1)} \geq \Delta_y^{(t)} + \alpha'\lambda_t/2 \geq \alpha'\lambda_t/2.$$

Recalling from Lemma 1 that $\lambda_t$, the norm of the prediction vectors, is as $(1+\gamma')^t$, we conclude that for all $t \geq t_0+1, \Delta_y^{(t)} \geq (1 + \gamma')^t$.

Finally, note that since $\Delta_t \geq (1 + \alpha')^t \Delta_0$,

$$t_0 \leq \inf\{t : \alpha\Delta_t \geq \log(4K)\} \leq \inf\{t : (1 + \alpha')^t \geq \log(4K)/\alpha\Delta_0\} \leq \frac{1}{\alpha'} \log\frac{\log 4K}{\alpha\Delta_0} + 1.$$

$\square$

# D  Additional insights into Mean-Shift Classification

## D.1  The role of $\alpha$

The hyperparameter $\alpha$ controls the influence data geometry exerts on the weights $A$. Recall that the data point updates take the form

$$\boldsymbol{x}_i^{(t+1)} = \boldsymbol{x}_i^{(t)} + \alpha' \sum_{j \in S_c} A_{ij}^{(c)} \boldsymbol{x}_j^{(t)}.$$

When $\alpha = 0$ the updates are solely driven by the relationships between labels, which are initialized to be intra-class orthogonal. At the same time however, this leads to the test point $x_{\text{test}}$ to be pulled "equally" towards the direction of all classes, which means it does not move from its initial position. Thus, $\alpha > 0$ plays an important role not only for the mean-shift dynamics of the train points but also for the trajectory of the probe point.

## D.2  Mean-shift classification on non-linear data distributions

In this section, we stress-test the extracted mean-shift classifier by applying it to a range of data distributions in two dimensions that go beyond the linear setting studied in the main text.

**Separated Ellipses**  We generate two classes of anisotropic Gaussian clusters. Each class $k \in \{0, 1\}$ consists of points sampled from a multivariate normal distribution $\mathcal{N}(\boldsymbol{\mu}_k, \Sigma)$, where $\boldsymbol{\mu}_0 = (-s, -s)$ and $\boldsymbol{\mu}_1 = (s, s)$ provide linear separation.

**Voronoi Cells**  We simulate a multi-class problem by defining $K$ random centroids $\{\mathbf{c}_k\}_{k=1}^K$ within a bounded region $[-B, B]^2$. Data points $\mathbf{x}_i$ are sampled uniformly from this region. The label $y_i$ for each point is assigned according to the nearest centroid, effectively partitioning the space into Voronoi cells: $y_i = \operatorname{argmin}_k \|\mathbf{x}_i - \mathbf{c}_k\|_2$.

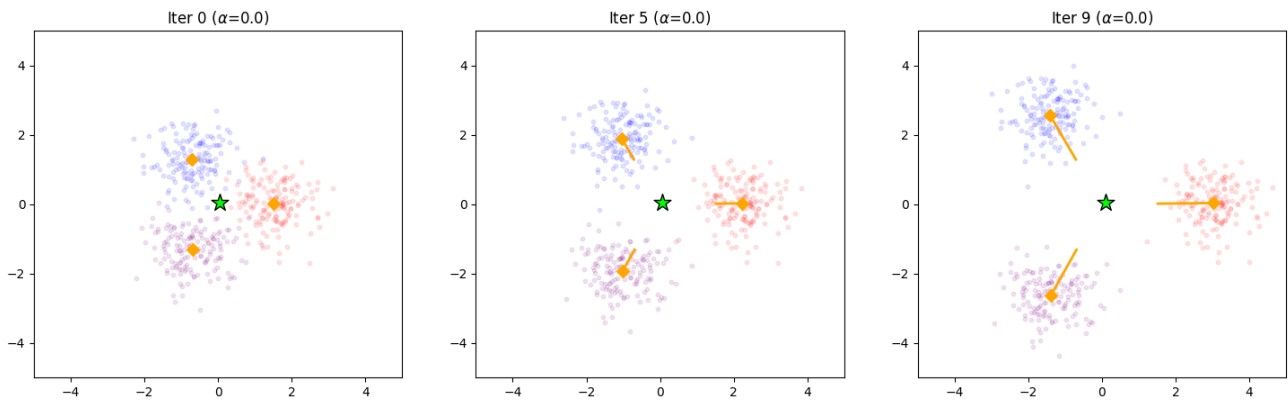

*Figure 18.* Setting $\alpha = 0$ gives us label-driven mean-shift, and the probe point does not move.

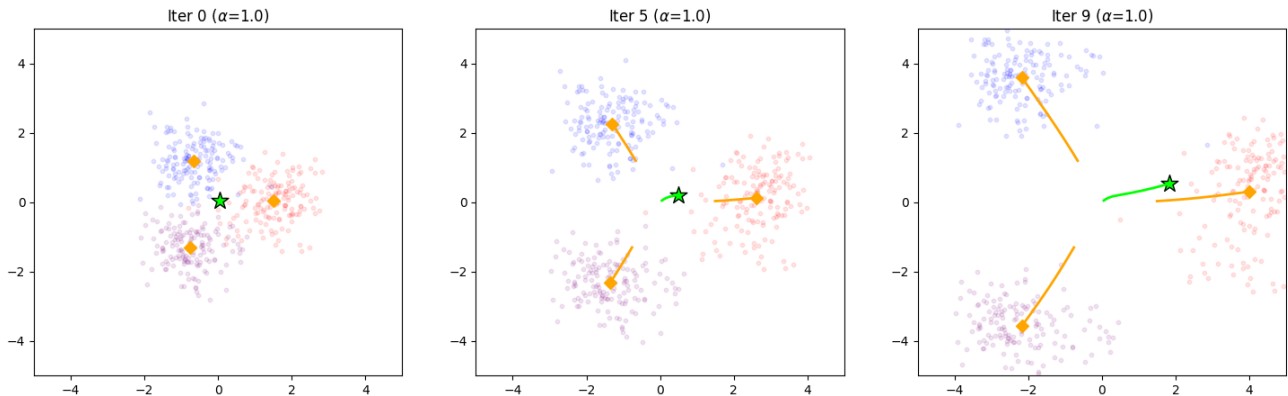

*Figure 19.* Setting $\alpha > 0$ leads to a probe point trajectory.

**Concentric Circles**    This dataset consists of $K$ classes arranged as nested rings. For the $k$-th class, points are generated using polar coordinates where the radius is fixed at $r_k = 1 + k$ and the angle $\theta$ is sampled uniformly from $[0, 2\pi)$. Gaussian noise $\epsilon \sim \mathcal{N}(0, \sigma^2 I)$ is added to the Cartesian coordinates $(r_k \cos \theta, r_k \sin \theta)$ to create diffuse ring structures that are not linearly separable. Surprisingly, the mean-shift classifier still separates these data well despite the inherently non-linear decision boundaries.

**Intertwined Spirals**    Two classes are generated as interleaving spiral arms. For each class $k \in \{0, 1\}$, we define a parametric curve where the radius $r$ grows linearly with the angle $\theta \in [0, 4\pi]$. The coordinates are given by $\mathbf{x} = (\theta \cos(\theta + \phi_k), \theta \sin(\theta + \phi_k))$, where $\phi_k$ is a phase offset (e.g., 0 and $\pi$). Gaussian noise is added to the points, creating a highly non-convex geometry that is difficult to separate with simple distance metrics. We did not find any parameter setting for which the mean-shift classifier reliably separates these class boundaries, and so this is a failure mode for these dynamics. Concretely, it appears that for this data, the strongly intertwined geometry overwhelms the label-separations, which are further hampered by the fact that the cluster-centroids for the two labels are essentially coincident.

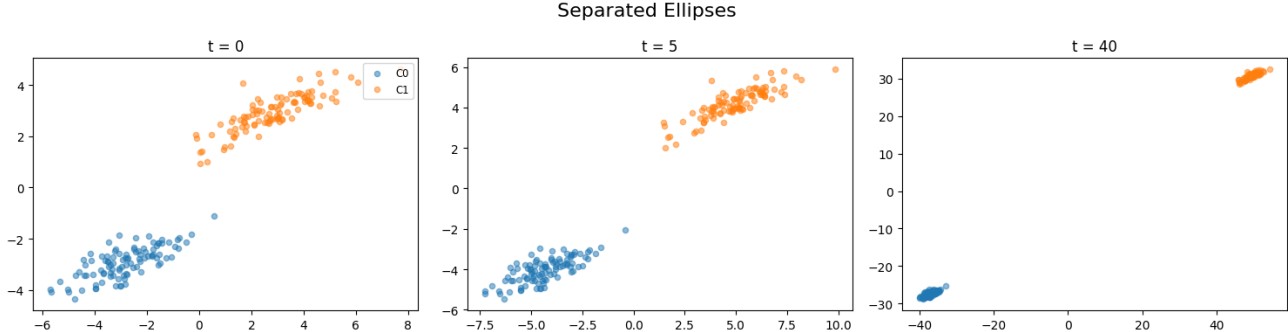

*Figure 20.* Mean-Shift Effects on separated ellipses.

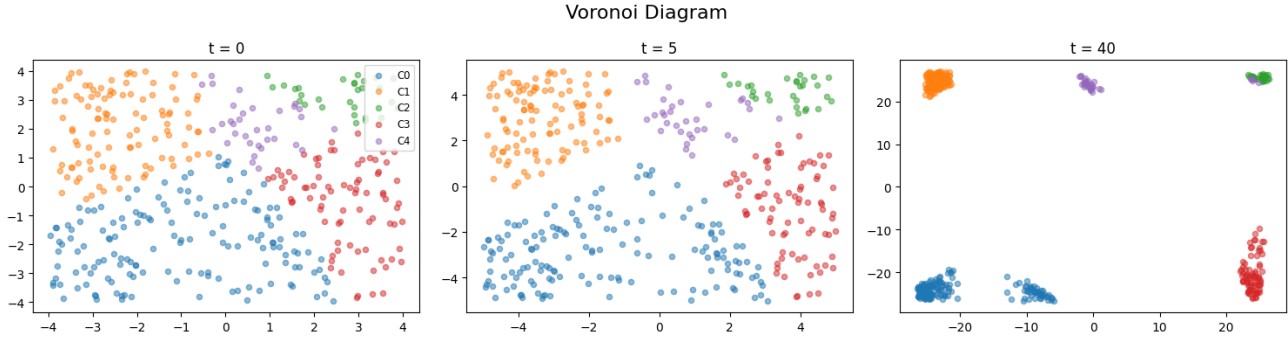

*Figure 21.* Voronoi Cells.

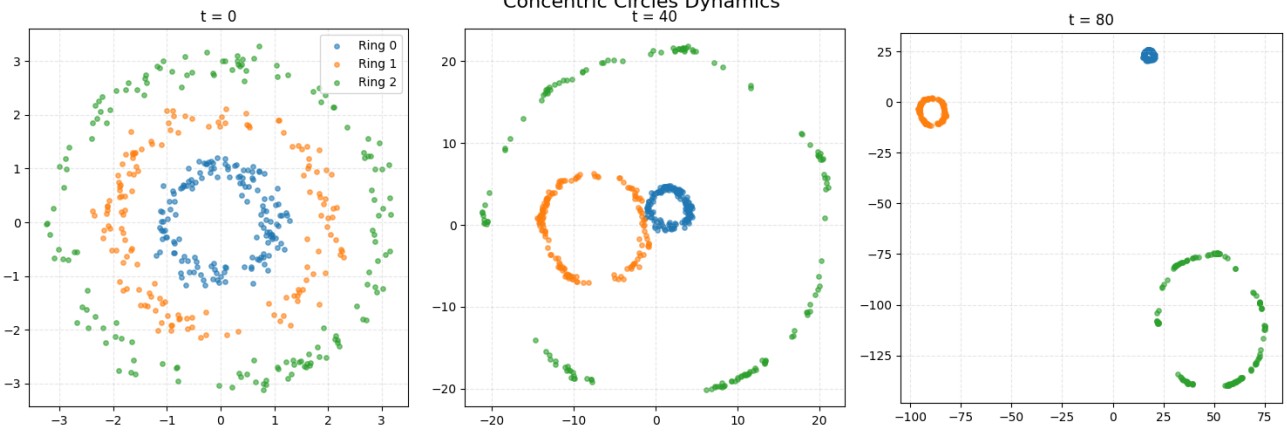

*Figure 22.* Concentric Circles

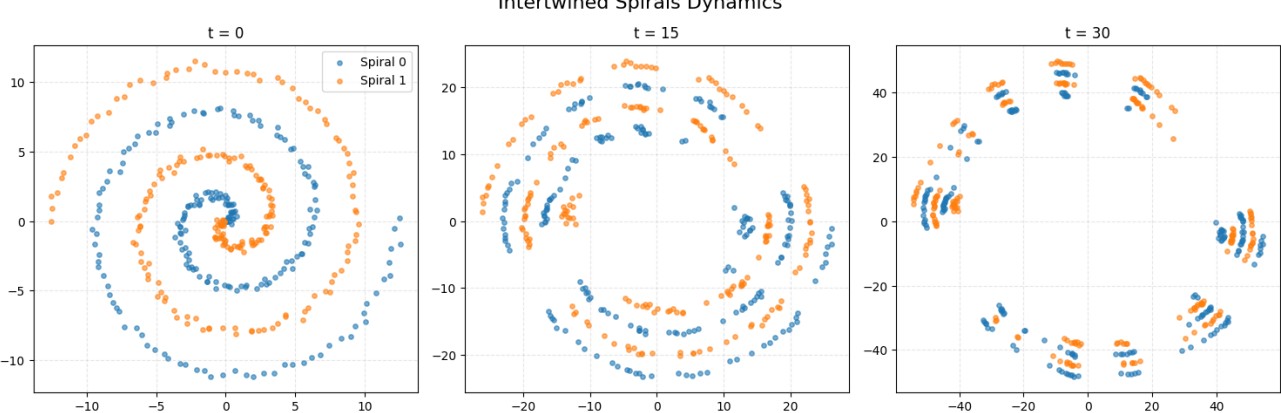

*Figure 23.* Spirals Mean-Shift

