# OpenReview forum: "Symmetry Reveals Layerwise Dynamics: How Transformers Perform In-Context Classification"
_ICML.cc/2026/Conference — ICML 2026 spotlight_

### Official Review · Reviewer_wU45 · 2026-03-13

**Soundness:** 3
**Presentation:** 3
**Significance:** 3
**Originality:** 3
**Overall Recommendation:** 4
**Confidence:** 3

**Summary:**

The authors studyy the effect of enforcing feature and label permutation invariance on training a Transformer to solve a in-context classification task. The contributions are as follows:
1. Authors propose a symmetry-enforced training scheme, by explicitly permuting the input to each attention layer, and then un-permuting it after the attention output.
2. The authors use an explicit recursion (mean-shift algorithm) to characterize the Transformer's inference time computation.
3. The authors experimentally validate that the Transformer weights are as predicted after training.
4. The authors establish the convergence of the proposed algorithm in the linear classification setting.
5. The authors empirically validate that their learned transformer performs competitively with known algorithm baselines, in the supervised and semi-supervised settings.

**Compliance With Llm Reviewing Policy:**

Affirmed.

**Final Justification:**

my concerns were addressed.

**Key Questions For Authors:**

## Questions
1. Is the un-permutation operation necessary after each attention layer?
2. Why is it important to randomize the permutation per layer, instead of randomizing the permutation, say, per forward call (i.e. apply random permutation only to the input)
3. How does this "forced permutation" relate to more practical methods, such as random rotation/crop in training vision models?
4. Why did Figure 3 not compare the accuracy of the symmetry-enforced transformer versus standard transformer?
5. Is the weight matrix (6) not exactly the proposed GD construction in von Oswald et al 2023, as well as in many of the follow-up papers to it? Is it because the mean-shift algorithm in the linear setting includes gradient descent as a special case?
6. In (6), why do authors assume that the bottom-right quadrant of W is $\gamma I_K$? The bottom right quadrant of W, based on Figure 2, appears to be non-diagonal?
7. What is the training loss in section 4? Is it cross-entropy loss of logistic classification?

**Strengths And Weaknesses:**

## Strengths
1. The experiment in Figure 2 (and 7) clearly visualizes the difference between the permutation-invariance-enforced training scheme and the default training scheme, showing that  the proposed method produces more interpretable weights.
2. The coupled-mean-shift algorithm is possibly more natural than the GD perspective, because in GD there is a clear asymmetry in how the features and labels are handelled (often enforced by artifical sparsity constraints on attention weights.)
3. The applicability of the proposed method to semi-supervised learning is quite interesting. There are a few existing papers which look into how transformers can learn in context in the semi or un-supervised setting. A comparison between the mean-shift algorithm and these existing works would be helpful.

## Weakness
1. The feature-permutation invariance is a very strong assumption. As an example, if the input token is an image, this assumption says that any permutation of pixels is the same image. It is difficult to see why this assumption should be meaningful in a practical application.
2. Given that permutation invariance is usually not enforced in practice, it is unclear how applicable the findings of this paper are to a real Transformer, especially if the problem does not naturally have feature permutation-invariance.
3. Theorem 4.1 only applies to the linear classification problem.
4. Minor point that does not affect my score: I find the language to be unnecessarily complex at several places. For example: "trained transformers repeatedly instantiate a shared dynamical template: a label-aware, geometry-driven mean-shift motif. This motif is not merely a post-hoc analogy...". What does "instantiate ... shared dynamical template: ... motif" mean? If this came from LLM polishing, I suggest the authors ask the model to use more straight-forward language.

---

> ### Author Rebuttal · Authors · 2026-03-28
>
> Thank you for the careful and constructive review. We are glad the reviewer found the weight visualizations, the mean-shift perspective, and the semi-supervised extension interesting. We also agree that parts of the prose can be simplified, and we will revise the wording accordingly. We first speak to the weaknesses you raised.
>
> **1. Practical relevance of feature-permutation symmetry.** Yes, arbitrary feature permutations are not natural in images, and we do not claim pixel permutations should be enforced in vision. In our controlled prediction tasks, feature/label/example permutations are task symmetries; enforcing them removes gauge freedom and makes the learned inference procedure identifiable. We introduce the symmetry operation as an interpretability tool. Unlike random crops or rotations, which augment inputs, our permutation sandwich is a layer-wise constraint that forces computation to commute with the symmetry. Analogously in vision, we could enforce translation or rotation equivariances with the same goal of improving identifiability. Moreover, we want to highlight that **feature permutation invariances can arise naturally in practice**. For instance, models with learned token embeddings have an exact latent coordinate relabeling symmetry: consistently permuting embedding dimensions is a pure reparameterization.
>
> **2. Applicability to real transformers due to permutation invariance.** We agree that this is a controlled mechanistic study, not a direct claim about all naturally trained transformers. See also our response to reviewer CfPs. Our argument proceeds in two steps: first, we show that the symmetrized and unconstrained transformers remain closely aligned in which examples they rely on, in pointwise predicted probabilities, and in aggregate accuracy across problem variations (Fig. 4; Appx. B.1 / Fig. 10). Second, symmetrization induces weight structure that lets us extract Eq. (7). In that sense, symmetry-enforced training is a methodological tool for exposing the computation learned by the unconstrained transformer. We will sharpen this exposition in the paper.
>
> **3. Scope of the formal theorem.** Yes, the formal guarantee is for linear multiclass. The paper first extracts dynamics from the trained transformer, then analyzes them in the core linear regime. The nonlinear/Voronoi discussion is motivated by the mechanism and then validated empirically, not proved formally. We will make that theory/empirics boundary much more explicit.
>
> **Relation to prior GD-style ICL work (Q5).** While the block form in Eq. (6) indeed is similar to von Oswald et al. (and other prior observations), notice that we are studying classification problems with a softmax-activation transformer, while this prior work was on linear regression with linear-activation attention blocks. This makes the resulting behavior of the derived dynamics (7) very different than what was seen in such prior work. We will underscore this point in the text. More fundamentally, this family (and mean-shift in particular) is not GD: the core computation is a mixing/expansive dynamic in data/representation space rather than parameter-space ERM. We will also position the semi-supervised result more clearly relative to prior semi-/unsupervised ICL work.
>
> Questions:
>
> 1. Yes, un-permutation is necessary. The role of Eq. (4) is to conjugate each attention block by the symmetry action, so that the block is forced to commute with the symmetry while the residual stream remains in the original coordinate system. Without the inverse permutation, the representation would remain in a random basis, and the next layer / residual addition would no longer be operating in a shared coordinate system.
>
> 2. Notice that the data-distribution already has permutation symmetry. Thus, a fixed permutation at the beginning can just be absorbed into the data, and so would not give structured weights. By permuting layer-wise, we prevent the model from coordinating on a single hidden basis, and forces each layer to explicitly process inter-point relations rather than absolute positions. We will explain this more directly.
>
> 3. See (1): unlike crops/rotations, our intervention is an internal layerwise symmetry constraint, not input augmentation.
>
> 4. Fig. 3 benchmarks against standard baselines; the direct unconstrained-vs-symmetrized comparison is in Figs. 4 and 10, where the two models track each other closely.
>
> 5. See above.
>
> 6. We do not claim that the raw learned bottom-right block is exactly $\gamma I_K$: Eq. (6) keeps the dominant diagonal term and drops a weaker shared-background / low-rank component visible in Fig. 2, and Appx. B.1 shows that this simplification remains behaviorally faithful: coarse clustering already preserves essentially all predictive behavior, and the simplified abstraction remains closely aligned with the original model. We will highlight this abstraction step in our revision.
>
> 7. Standard multiclass cross-entropy on the query label.

---

> > ### Author Rebuttal · Reviewer_wU45 · 2026-04-03
> >
> > I will maintain my score.

---

### Official Review · Reviewer_1WGR · 2026-03-13

**Soundness:** 3
**Presentation:** 3
**Significance:** 3
**Originality:** 3
**Overall Recommendation:** 5
**Confidence:** 3

**Summary:**

This paper investigates the inference-time mechanics of in-context learning by enforcing layerwise task symmetries to make internal computations identifiable. The authors discover that transformers consistently implement a shared algorithmic motif of label-aware mean-shift dynamics, rather than implicit gradient descent, to iteratively amplify geometric class separation. This extracted recursion successfully explains transformer performance across diverse classification settings, including non-linear Voronoi tasks and semi-supervised learning.

**Compliance With Llm Reviewing Policy:**

Affirmed.

**Key Questions For Authors:**

see weaknesses

**Limitations:**

yes

**Strengths And Weaknesses:**

Strengths:
- The paper utilizes novel techniques to train algorithmically interpretable transformers and extract their internal mechanisms.They identify a novel mechanistic motif, label-aware mean-shift, that is distinct from e.g., in-context gradient descent that is widely discussed in the literature. These results contribute to the fundamental understanding of transformer operations and can inspire further theoretical work to go beyond analyzing in-context GD-like algorithms
- The discovered mechanism regarding how transformers process inputs in representation space is particularly interesting as it offers a new perspective on internal hidden-state dynamics.
- experiments demonstrate that this mean-shift motif emerges consistently across various settings, including linear classification.

Weaknesses:

These factors: the reliance on the enforcement of layer-wise symmetry, the fact that the discovered mechanism has clear failure modes under high nonlinearity, and the sensitivity to the hyperparameter $\alpha$, leave it unclear how much the results contribute to the understanding of naturally trained transformers operating on complex, real-world data

---

> ### Author Rebuttal · Authors · 2026-03-28
>
> We thank the reviewer for the constructive assessment and for highlighting the novelty of the label-aware mean-shift motif. We agree that the main limitation is how far a symmetry-first analysis in a controlled setting can speak to naturally trained transformers on complex real-world data, and we will sharpen that scope in the revision.
>
> **1. The role of layer-wise symmetry**
>
>   We introduce layer-wise symmetrization as a **diagnostic intervention** that renders inference identifiable in a controlled softmax-attention ICL setting. In this regime, symmetrization removes symmetry-breaking degrees of freedom while preserving the underlying algorithm implemented by the original, unconstrained model. This makes it possible to recover a simpler yet faithful surrogate model that yields mechanistic insight into the original transformer. The key assumption is only that the prediction problem exhibits some symmetries. For instance, any model that represents discrete tokens via learned embeddings has an exact **latent coordinate relabeling symmetry** of the embedding space: permuting embedding dimensions consistently is a pure reparameterization. Our intervention can be applied whenever the prediction problem approximately or exactly respects such symmetries.
>
> **2. Failure modes under high nonlinearity**
>
> Our main results are about three trained settings: linear classification, Voronoi classification, and semi-supervised linear classification. The experiments in the appendix on concentric circles and intertwined spirals are **purely exploratory**: we do not train transformers on those tasks, but only test whether the same extracted mean-shift iteration transfers directly. It is therefore notable that this mechanism works well even on concentric circles, but not on intertwined spirals. We do not interpret the latter as showing that a transformer cannot solve spirals; only that the specific mechanism identified in our main settings does not transfer unchanged. These experiments are included only to probe the scope of the mean-shift mechanism, not to qualify the main identification result on our three key tasks. We will make this distinction much clearer in the final revision.
>
> **3. The role of $\alpha$**
>
>   We will clarify that $\alpha$ plays a mechanistic role: it controls how strongly feature geometry enters the attention scores. In the fixed-parameter analysis, the relevant condition is on the product $\alpha \Delta_0$, not on $\alpha$ in isolation, so we should not phrase the result as unconditional insensitivity to $\alpha$. In the trained transformers, moreover, the mechanism is governed by a learned **layerwise schedule** rather than a single hand-set scalar.
>
> **4. Applicability to real-world architecture and data**
>
> This is an important scope question, and it was also raised by reviewer CfPs. We address it in detail in point 4 of our response to reviewer CfPs, and refer the reader there for a fuller discussion.

---

> > ### Author Rebuttal · Reviewer_1WGR · 2026-04-03
> >
> > Thanks for the response. I am satisfied with the authors’ responses and will keep my positive score.

---

### Official Review · Reviewer_xYXX · 2026-03-17

**Soundness:** 2
**Presentation:** 2
**Significance:** 2
**Originality:** 3
**Overall Recommendation:** 4
**Confidence:** 4

**Summary:**

This paper studies in-context classification by enforcing the task’s natural symmetries inside the network, with the goal of making the transformer’s internal computation identifiable.
The authors focus on a single-head, attention-only softmax transformer and explicitly omit MLPs, layer norm, and multi-head attention to keep the mechanism tractable.
They then introduce layerwise randomized permutations to enforce feature/label symmetries and use the resulting structured model to extract a low-dimensional recursion that they interpret as a label-aware mean-shift classifier.

**Compliance With Llm Reviewing Policy:**

Affirmed.

**Final Justification:**

My concerns have been resolved, so I raise my score.

**Key Questions For Authors:**

1. Could the authors clarify the exact logical relationship between Theorem C.1 and Lemma 3?
 Lemma 3 appears to require stronger pointwise assumptions than Theorem C.1 proves.
I would strongly encourage the authors either to prove an additional lemma that derives those pointwise conditions from C.1 under extra assumptions, or explicitly state Lemma 3’s training-separation condition as an additional assumption rather than phrasing it as if it follows from C.1.

2. How sensitive is the theory to class imbalance and weak initial margins?
Since Theorem C.2 explicitly uses balanced classes and a strong initial condition on  alpha Delta_0, I would like to see results under class imbalance and smaller initial geometric margins.

3. Could the semi-supervised results be repeated with a single model across multiple context lengths?
The current setup trains one model per context length. A more convincing demonstration would fix one trained model and vary only the amount of unlabeled context at test time.

 If these points were cleaned up, my score would increase.

**Limitations:**

See Weeknesses and Questions.

**Strengths And Weaknesses:**

Strengths
1. Using symmetry not just as an invariance principle but as a mechanism-extraction tool is  novel.
The paper turns an interpretability problem into an identifiability problem, which is a conceptual contribution.
2. The move from an unconstrained transformer to an explicit recursion, and then to a label-aware mean-shift interpretation, is interesting and easy to reason about.
3. The paper connects the extracted update rule to supervised clustering and prototype formation.

Weaknesses
1. The bridge from Theorem C.1 to Lemma 3 is the main theoretical weakness.
    Theorem C.1 proves growth of a centroid-level global directional margin; however, Lemma 3 assumes much stronger pointwise intra class and inter-class inner-product conditions, stated as "By Theorem C.1, let us assume …".  Those assumptions are not implied by Theorem C.1 as stated, and they are used essentially in the proof of the test-point recursion. This means the paper’s test-point theory does not cleanly follow from the preceding theorem; instead, it relies on an additional stronger assumption.

2. The main theorem relies on strong conditions.
Theorem C.2 assumes a nontrivial initial geometric margin, a sufficiently large product alpha Delta_0, and uses a balanced-class simplification in the logit-gap bound. This makes the result a stylized sufficient-condition theorem rather than a broad explanation of when in-context classification should work.

3. In the semi-supervised section, the paper trains a separate model for each context length. That is useful for estimating best-case behavior, but it leaves open how much of the gain survives in a single model evaluated across varying prompt lengths.

4. The paper’s own appendix shows a failure mode on intertwined spirals. That is not fatal, but it suggests the mean-shift explanation has a narrower domain of validity than the main narrative may initially suggest.

---

> ### Author Rebuttal · Authors · 2026-03-26
>
> **1. On Lemma 3.**
>
> Thanks for your precise reading. The pointer to Thm. C.1 indeed does require clarification. C.1 is a centroid-level directional-margin statement, whereas Lemma 3 uses stronger pointwise train/test separation conditions. A clearer way to state the argument is to separate the train and test margins explicitly.
>
> Let $$ R\_t = \min\_{i \in S\_{c^\*}} \langle x^t, x\_i^t\rangle, L\_t = \max\_{i \not\in S\_{c^\*}} \langle x^t, x\_i^t\rangle,  $$
> and similarly for training points $$ \rho\_t = \min_{i,j \in S\_{c^\*}} \langle x\_i^t, x\_j^t\rangle, \lambda\_t = \max_{i \in S\_{c^\*}, j \not \in S\_{c^*}} \langle x\_i^t, x\_j^t\rangle.$$
>
> Finally, introduce the training margin $\Gamma\_t := \rho\_t - \lambda\_t,$ and the test margin $\Delta\_t := R\_t - L\_t$. Then, with only cosmetic changes, **the proof as written establishes** $$ \Gamma\_{t+1} \ge (1+\alpha')^2 \Gamma\_t, \textrm{ and } \Delta\_{t+1} \ge (1+\alpha')\Delta\_t + \alpha'(1+\alpha')p\_{c^\*}^t \Gamma\_t - 2\alpha' (1-p\_{c^\*}^t) (1+\alpha')^{2t+1}. $$
>
> Naturally, then, $\tilde{\Delta}\_t := \min(\Delta\_t, \Gamma\_t)$ satisfies the inequality in Lemma 3, and so we can get the same result, but with the condition on $\tilde\Delta\_0 = \min(\Delta\_0, \Gamma\_0)$ instead of on $\Delta\_0$. Thereafter, **Theorem 4.1 follows**, with condition $$ \alpha \Delta\_0 \ge \log(K) + \log(1 + 2/\tilde{\Delta}\_0).$$ Note that the case where $\Delta\_0 < \Gamma\_0$ (i.e., the test margin is smaller than train margin) is the more interesting case here.
>
> *Bonus.* But in fact **even more is true.** If we incorporate the above bounds into the calculation starting line 1200 in the proof of Thm. 4.1, then it turns out that the growth $\Delta\_t \ge (1+\alpha')\Delta\_t$ is sustained so long as for all $t$, $p\_{c^*}^t (\Gamma\_0 + 1) \ge 1$. This simplifies lines 1214 to 1267 considerably, and gives this growth so long as $\alpha \Delta\_0 \ge \log(K) + \log(1 + 1/\Gamma\_0)$. Then Thm. 4.1 follows with this condition instead.
>
> *Overall.* We had actually intended to define $R\_t, L\_t$ as the minimum of the above $R\_t$ and $\rho\_t$, but made last-minute changes to clean up the presentation of the result in the main text, which introduced this issue. The reference to C.1 was only supposed to point to the fact that margins increases. Nevertheless, it actually comes together better through keeping the test and train margins separate :)
>
> ---
>
> **2. On class-balance and margin in the theory.**
>
> Class balance is not an issue at all. If we define $m =  \min_{c} |S\_{c^\*}|/|S_c|,$ then we can incorporate any $m > 0$ in the theory. The only change is that the bound in line 1194 has to be relaxed to $(K/m) e^{-\alpha \Delta\_t}.$ Thereafter, everything remains the same, except all instances of $K$ turn into $K/m$.
>
> On the other hand, the positive margin is indeed required in the theory, although any arbitrarily small margin is enough (for large enough $\alpha$). As such you are correct in noting that this is only a sufficient condition. In fact, note that we only analyze a constant step-size iteration, whereas the transformer can and does choose a more complex schedule of these parameters. Our intention here was to illustrate how this family of dynamics can form a good prediction. Nailing down every aspect of this family of iterative methods should be much harder though. To illustrate this difficulty, we point out that even the mean-shift algorithm itself is not completely understood outside of somewhat specialized cases.
>
> ---
>
> **3. Semi-supervised in one transformer**
>
> Thanks for the suggestion---in a nutshell, we did this and it works well!
>
> To address this concern directly, we took your suggestion and trained a **single** transformer while randomizing the number of unlabeled context points across batches during training, and then evaluated that fixed model across different context sizes. The qualitative effect persists: as more unlabeled points are added, the same transformer improves substantially, while the supervised baselines remain flat and the graph-based SSL baselines remain below the transformer. Thus, the SSL effect is not an artifact of retraining a separate model for each context length; a single model can exploit additional unlabeled geometry at test time.
>
> > The plot is attached at this anonymized link: https://imgur.com/a/QkdB5oa
>
> We will include it in revision; clarify that previous per-context is to be interpreted as an oracle upper-bound study; this new result shows same phenomenon for fixed-model.
>
> ---
>
> **4. Domain validity**
>
> We agree and will make it explicit (see 1WGR response). The intertwined-spirals example in appendix is a failure mode: it shows that the mean-shift mechanism has a broad but non-universal domain of validity. Our claim is that the extracted recursion explains a geometry-driven regime, including linear/Voronoi/SSL settings, and not that it is the correct description of all nonlinear ICL classification tasks.

---

> > ### Author Rebuttal · Reviewer_xYXX · 2026-04-03
> >
> > Thank you for your response. My concerns have been resolved, so I will raise my score.

---

### Official Review · Reviewer_CfPs · 2026-03-19

**Soundness:** 3
**Presentation:** 3
**Significance:** 3
**Originality:** 3
**Overall Recommendation:** 5
**Confidence:** 2

**Summary:**

This paper tackles the problem of making in-context classification in transformers more identifiable by enforcing symmetries at each layer. The enforces symmetry is used to extract the emergent algorithm which suggests  1) coupled mean-shift dynamics mechanism is uncovered 2) ICL behavior can be explained through shared algorithmic motif that is not gradient descent.

**Compliance With Llm Reviewing Policy:**

Affirmed.

**Final Justification:**

My concerns have been answered so I am raising my score.

**Key Questions For Authors:**

1. How does the number of classes affect the aspect of label symmetries?
2. Line 214: Interpretation via behavioral alignment. Can you please point to the results supporting this?
3. Section 4.1: Can you provide more details on how equation 6 is derived and the intuition behind using this abstraction?
4. How do the observations using a simple transformer architecture apply to real-world architectures?

**Limitations:**

The paper uses a simple architecture and synthetic dataset. A discussion on how do the observations in this work translate to real-world models and datasets will be valuable.

**Strengths And Weaknesses:**

Strengths:
1. The paper tackles an important fundamental problem of understanding how the ICL algorithm works through strong theoretical and empirical results.
2. The contribution and the idea of enforcing symmetry to analyze the underlying algorithm are novel

Weaknesses
1. The evaluation is conducted using a synthetic dataset, and how well it translates to the real dataset is not clear.
2. The transformer architecture explored here is very simple.

---

> ### Author Rebuttal · Authors · 2026-03-28
>
> We thank the reviewer for recognizing both the importance of the mechanistic question and the novelty of using symmetry to analyze the learned ICL algorithm.  We address points raised below.
>
> **1. How does the number of classes affect the aspect of label symmetries?**
>
> The relevant symmetry on labels is the permutation group over the $K$ classes. Changing $K$ changes the size of the symmetry group and the label-subspace dimension, but not the symmetry itself: relabeling classes induces the corresponding relabeling of the prediction. This is the symmetry formalized in the paper, where label symmetries are fixed permutations of all $y_i$ in the prompt. The extracted recursion is written directly in terms of $Y \in \mathbb{R}^{(n+1)\times K}$, so its form naturally scales with $K$. In the label-dominated regime, the analysis shows that labels remain class-aligned and the training attention becomes increasingly class-structured, which is the source of the mean-shift behavior. We will clarify this explicitly in the final paper.
>
> **2. Line 214: “Interpretation via behavioral alignment.” Can you point to the results supporting this?**
>
> By “behavioral alignment,” we mean that symmetrization removes symmetry-breaking coordinate freedom without changing the underlying inference-time algorithm. The paper supports this claim in three complementary ways:
> (1) Fig 4 (left): the unconstrained and symmetrized models rely on essentially the same in-context examples, as shown by gradient-based influence alignment $(\text{Spearman's } \rho \approx 0.8)$.
> (2) Fig 4 (right): their predictions agree pointwise on the same prompts: the predicted class probabilities are nearly identical instance-by-instance ($R^2 = 0.964$ across 2,000 instances).
> (3) Fig 10: agreement persists across broader sweeps of depth, context length, dimension, and margin in Appendix B.1.
> We agree this should be more explicit and we will revise the paper to refer to these results when we first mention “behavioral alignment” on L214.
>
> **3. Section 4.1: Can you provide more details on Eq. (6)?**
>
> Eq. (6) is not assumed a priori; it is abstracted from the learned structure of the symmetrized transformer weights (Fig 2, right). After enforcing symmetry, the weight matrices collapse to a much more regular structure, and the dominant pattern is a block-diagonal form with one scalar acting on feature coordinates and one scalar acting on label coordinates. The quantitative support for this abstraction is given in Appendix B.1: coarse clustering of each weight matrix preserves essentially all predictive behavior, and the simplified abstraction remains behaviorally faithful to the original model (Fig 11). Intuitively, $\alpha$ captures the feature-similarity contribution to attention, while $\gamma$ captures the label-similarity contribution; the abstraction isolates exactly those two dominant degrees of freedom that become visible after symmetrization. We will expand Section 4.1 to make this derivation more explicit and connect Eq. (6) more clearly to both Fig. 2 and the faithfulness results in Appendix B.1.
>
> **4. How do the observations using a simple transformer architecture apply to real-world architectures?**
>
> This is an important scope question. Our goal in this paper is a strong mechanistic claim, i.e., to identify the computation implemented by the model. That level of end-to-end identification is currently feasible only in controlled settings. For this reason, the paper explicitly studies attention-only transformers with softmax and omits MLPs in order to keep the mechanism identifiable while still retaining data-dependent softmax attention; while simpler than full modern architectures, is goes beyond linear-attention analyses that remove softmax. This is the next step in a long line of mechanistic ICL work: prior papers identified optimizer-like behavior mainly in linear-attention transformers trained on regression, whereas our contribution moves to softmax transformers and a multiclass classification task, and extracts a different mechanism; a representation-space, label-aware mean-shift dynamic rather than parameter-space GD.
>
> We use synthetic task families because train-from-scratch ICL requires many tasks drawn from a common, known distribution, and synthetic tasks provide a controlled and reproducible testbed for that objective. Our claim is not that the present paper fully characterizes naturally trained large-scale transformers on real-world data. Rather, we view the contribution as isolating a symmetry-constrained algorithmic motif in a regime where it can be extracted exactly, which can then guide future investigation in richer architectures and domains. This scope is already partially acknowledged in the conclusion, where we say that the current method focuses on synthetic tasks with evident symmetries and that extending it to mechanisms not aligned with layer-wise symmetries is future work. We will sharpen this positioning in the final paper.

---

> > ### Author Rebuttal · Reviewer_CfPs · 2026-04-05
> >
> > Thanks for the rebuttal. My questions have been answered, and I will raise my score.

---

### Decision · Program_Chairs · 2026-04-30

**Decision:**

Accept (spotlight)

**Comment:**

This submission investigates the inference-time mechanism of in-context learning by enforcing layerwise task symmetries to make internal computations identifiable.

All the reviewers acknowledged this contribution for providing a new perspective on in-context learning in attention-based models. During the rebuttal period, the authors addressed the reviewers' concerns well regarding the assumptions of the theorems and scope issues (how it connects with real transformers and how it differs from the GD-based in-context learning mechanism).

Therefore, I recommend a clear acceptance, and suggest the authors include the key concerns and discussions during the review period to the final version of the paper.